

# Late Holocene channel pattern change from laterally stable to meandering caused by climate and land use changes

Jasper H.J. Candel[1], Maarten G. Kleinhans[2], Bart Makaske[1], Wim Z. Hoek[2], Cindy Quik[1], Jakob Wallinga[1]

[1]Soil Geography and Landscape Group, Wageningen University & Research, Wageningen, PO Box 47, 6700AA, The Netherlands
[2]Department of Physical Geography, Utrecht University, Utrecht, PO Box 80125, 3508TC, The Netherlands

*Correspondence to*: Jasper H.J. Candel (jasper.candel@wur.nl)

**Abstract.** River channel patterns may alter due to changes in hydrological regime, related to changes in climate or land cover. Such changes are well documented for transitions between meandering and braiding rivers, whereas channel pattern changes between laterally stable and meandering rivers are poorly documented and understood. We identified a river that was laterally almost stable throughout the Holocene until the Late Middle Ages, after which large meanders formed at lateral migration rates of about 2 m yr$^{-1}$. The lateral stability before the Late Middle Ages was proven using a combination of coring information, ground-penetrating radar (GPR), radiocarbon ($^{14}$C) dating, and optically stimulated luminescence (OSL) dating. Our objective of this work is to identify the possible causes for the meander initiation. We carried out a unique reconstruction of bankfull discharge as a function of time, based on channel dimensions that were reconstructed from the scroll bar sequence using coring information and GPR data, combined with chronological constraints from historical maps and OSL dating. Empirical channel and bar pattern models were used to determine the potential for meandering and to identify the causes of meander initiation. Several potential causes were investigated, varying from discharge regime changes to increased sediment input. Our investigation shows that bankfull discharge was two to five times higher during the meandering phase compared to the laterally stable phase. This increase likely reflects climate changes related to the Little Ice Age and land use changes in the catchment, in particular as a result of peat reclamation and exploitation. We hypothesize that many low-energy meandering rivers were laterally stable during most of the Holocene, reflecting relatively low peak discharges during a stable climate and with limited human impact. However, channel deposits associated with such stable phases are poorly preserved, due to recent increase in dynamics of such systems. Considering the importance of climate and land use changes on the river channel pattern, successful river restoration requires an integral approach that includes scenarios of climate and land use changes in the catchment.

## 1. Introduction

Channel patterns describe the planform of a river, which reflects the interaction of the river channel with its floodplain. Several channel patterns are classically distinguished. Laterally inactive channels consist of straight and stable sinuous



planforms, whereas laterally active channels consist of meandering and braiding planforms (Leopold and Wolman, 1957; Nanson and Knighton, 1996). Flume experiments and field data have shown that the channel pattern depends on several variables. Firstly, on the available potential specific stream power, which is the product of the channel-forming discharge and valley slope (Nanson and Croke, 1992; Van den Berg, 1995; Kleinhans and Van den Berg, 2011). Secondly, on the bank erodibility (Friedkin, 1945; Ferguson, 1987; Millar, 2000), which is determined by the presence of hard-rock in the valley

side (Turowski et al., 2008), the cohesiveness of the banks (Peakall et al., 2007), and by vegetation that can increase the bank strength (Millar, 2000; Gurnell, 2014). The ratio between bank strength and stream power eventually determines the channel pattern (Kleinhans, 2010).

Channel patterns can gradually change in response to environmental variations (Carson, 1984; Ferguson, 1987; Nanson and Croke, 1992; Notebaert and Verstraeten, 2010; Notebaert et al., 2018). Many examples of channel pattern changes from

braiding to meandering and vice versa are known to be associated with glacial/interglacial oscillations (Vandenberghe, 1995; Vandenberghe, 2002). Especially studies on the last glacial-interglacial transition have shown the simultaneous occurrence of channel pattern changes with a changing climate (e.g. Vandenberghe et al., 1994; Kasse et al., 2005; Kasse et al., 2016). Climate change affects the vegetation, sediment availability and discharge regime, and consequently the bank stability, sediment transport and potential specific stream power resulting in different channel patterns.

Within the Holocene, several examples are documented of channel pattern changes from braiding to meandering rivers and vice versa (Lewin et al., 1977; Passmore et al., 1993; Brewer and Lewin, 1998; Słowik, 2015). However, channel pattern changes between laterally stable and meandering rivers have rarely been reported (Lewin and Macklin, 2010). The exception is formed by human intervention, which transformed many meandering rivers into heavily regulated and laterally stable rivers by introducing weirs, dams, groynes and bank protection measures (Hesselink et al., 2003; Surian and Rinaldi, 2003;

Słowik, 2013; Hobo et al., 2014). Also the abandonment of former meandering valleys results in underfit, laterally stable rivers like the former Rhine branches in the Niers and Ijssel valley (Kasse et al., 2005; Janssens et al., 2012).

Both laterally stable and meandering rivers may display sinuous planforms, but the geomorphic processes in both rivers are different. Laterally stable channels are rivers witfhout meandering processes, i.e. helicoidal flows causing bar formation and bank erosion at a significant rate (Nanson and Knighton, 1996; Seminara, 2006; Kleinhans, 2010; Kleinhans and Van den

Berg, 2011; Candel et al., 2017). In fact, the bends and channel cut-offs in laterally stable rivers may be the result of random and local disturbances (e.g. falling trees, beavers, bank collapse after heavy rainfall, etc.) leading to very limited and local displacement of the channel. Meandering and laterally stable rivers should therefore be distinguished by their different patterns of bar and floodplain formation, rather than merely by planform (Kleinhans and Van den Berg, 2011; Candel et al., 2017).

Many studies have reported increased fluvial activity (e.g. increased discharge, sediment transport and deposition, and bank erosion rates) in relation to human, environmental and climatic pressures during the Holocene (e.g. Macklin, 1999; Bruneton et al., 2001; Kondolf et al., 2002; Gregory, 2006; Hoffmann et al., 2008; Lespez et al., 2008; Macklin et al., 2010; Notebaert and Verstraeten, 2010; Broothaerts et al., 2014; Kirchner et al., 2015; Lespez et al., 2015; Makaske et al., 2017; Notebaert et





al., 2018). An example of increased fluvial activity is known from the Pine Creek (Idaho, USA), where mining and deforestation combined with intensive grazing resulted in an increase of discharge and sediment input, followed by river widening and an increase in bank erosion (Kondolf et al., 2002). The reverse change has been observed in settings as a result of afforestation (Liébault and Piégay, 2001; Kondolf et al., 2002), or increase of riparian vegetation fixing the channel banks (Wolfert et al., 2001; Eekhout et al., 2014; Vargas-Luna et al., 2016).

A change in the hydrologic regime may invoke a change to a channel pattern associated with a higher energetic stage (Nanson and Croke, 1992). We conjecture that the change from laterally stable to meandering has occurred in some rivers for which increased Holocene fluvial activity was reported. The fact that such changes were not reported in the literature, may either mean that critical conditions for channel pattern change were not reached, or that evidence of such transitions is poorly preserved or left unnoticed. Notebaert and Verstraeten (2010) provided an extensive review of existing studies

concerning sediment accumulation in West and Central European river floodplains, and concluded sedimentation rates increased during the Middle and Late Holocene due to environmental changes. However, unknown is whether the channel pattern changed simultaneously with the floodplain, because channel deposits of the Early Holocene stable phase were unrecognized.

We suggest that identifying channel pattern changes requires more detailed historic accounts or a much higher resolution of

subsurface data than usually gathered, because palaeochannels of laterally stable rivers are poorly preserved in the fluvial archive of meandering channel belts (Van de Lageweg et al., 2016). Deposits and dimensions of channel reaches are not preserved when still active during the stable to meandering transition, because channel-belt dimensions increase. River reaches of laterally stable rivers can only be preserved when they are cut off by random and local disturbances prior to the meandering phase. Consequently, preservation potential of deposits associated to a laterally stable phase is very small, and

only channel reaches that have been subject to perturbations have a chance to be preserved. Using numeric (e.g. Oorschot et al., 2016) or scaled (e.g. Van Dijk et al., 2012) river simulation models is problematic for testing these ideas, because these have not yet been capable of reproducing channel pattern changes. This reflects the lack of understanding of river processes and patterns (Kleinhans, 2010), and the need to gather such information from field studies.

This research entails a case study focussing on a river where lateral activity during the past 500 to 600 years caused

spectacular meandering: the Overijsselse Vecht in The Netherlands (Fig. 1). Previous work on this system has identified a transition from braiding to meandering during the Late-Glacial (Huisink, 2000). In that study, and subsequent work, it was assumed that the river meandered throughout the Holocene until the river was channelized in 1914 AD (Huisink, 2000; Neefjes et al., 2011). However, Quik and Wallinga (submitted) reconstructed meander formation using a combination of optically stimulated luminescence (OSL) dating of scroll bars and planform reconstruction based on historical maps, and

found that the meanders were relatively young, with the oldest scroll bars dating from ca. 1400 AD. No fluvial deposits were found dating from before this period, except from a Holocene palaeochannel (here referred to as "Palaeochannel Q") in a ground-penetrating radar (GPR) profile recorded by Huisink (2000, p.123) 13 km upstream near Hardenberg (Fig. 1(b) and 2). Palaeochannel Q is relatively small compared to the meandering channel, and seems to lack scroll bars. Therefore, it is



5  questionable whether the Overijsselse Vecht meandered prior to 1400 AD. Alternatively, the river changed from a laterally stable into a meandering river in the Late Middle Ages. Our aims are (1) to identify whether a channel pattern change has occurred, by collecting and combining detailed subsurface and geochronological data, and (2) to identify causes for the exceptional lateral migration rates reported by Quik and Wallinga (submitted), and for the potential channel pattern change. Our study involves a high-resolution palaeohydrological reconstruction of the river prior to and during the pronounced

10  meandering phase to identify the potential causes.





**Figure 1: Maps of the Overijsselse Vecht. (a) Map showing the location of the Overijsselse Vecht catchment and the location of the study site. (b) Digital elevation map (DEM, Actueel Hoogtebestand Nederland, 0.5x0.5 m) (Van Heerd and Van't Zand, 1999) of the downstream section of the Overijsselse Vecht River, indicating both study sites: Junnerkoeland and Prathoek. DEM of the Junnerkoeland bend (c) and Prathoek bend (d), including locations of cores, OSL samples by Quik and Wallinga (submitted), the OSL and 14C samples from this study, the GPR transects, the grain size samples and inflection points. The possible historical course of Palaeochannel X according to Maas (1995) is indicated. (e) Zoomed-in figure of Palaeochannel X. (f) topographical military map (TMK) dating from 1852 AD (Van der Linden, 1973), showing the Overijsselse Vecht during its meandering phase.**



## 2. Study area

The Overijsselse Vecht (Fig. 1) is a low-energy, sand-bed river flowing from Germany into The Netherlands, with an average discharge ($Q_m$) and mean annual flood discharge ($Q_{maf}$) of 22.8 and 160 $m^3\ s^{-1}$, respectively, derived from the gauging station in Mariënberg for the period 1995 to 2015 (see location in Fig. 1(b)). The river has a length of 167 km, its catchment covers 3785 $km^2$ with the highest point +110 m above sea level (asl), and a relatively uniform valley slope of 1.42*10⁻⁴ in the Dutch part of its trajectory (Wolfert and Maas, 2007). The Overijsselse Vecht incised its current valley during the Late-Glacial within fluvioperiglacial sands, locally covered by aeolian coversands (Ter Wee, 1966; Huisink, 2000; Wolfert and Maas, 2007). During the Late Holocene, aeolian drift-sands formed along the Overijsselse Vecht as a result of agricultural overexploitation (Van Beek and Groenewoudt, 2011).

The Overijsselse Vecht is considered a classical example of the challenges of river restoration, due to the wide variety of stakeholders and interests in the area (Maas et al., 2007; Wolfert et al., 2009; Neefjes et al., 2011; Maas and Woestenburg, 2014). Water managers are struggling with the restoration of the Overijsselse Vecht in view of the meandering potential, the land use, recreational, groundwater and flood risk constraints (Damsté and Filius, pers. comm., September 16, 2016). The aim is to restore the river into a "half-natural lowland river", but the practical implementation of the restoration remains inconclusive. The Overijsselse Vecht was an actively meandering river until 1896, when weirs were constructed and parts of the river were channelized. The river was completely channelized in 1914 AD, with five weirs controlling the water levels. Recently, sinuous side channels bypassing the weirs have been created as part of river restoration aiming to restore past physical and ecological characteristics of the river.

At present the topography of the meandering phase is partly still intact in the floodplain (Maas, 1995). Wolfert and Maas (2007) reconstructed the pre-channelization planform from historical maps of 1720, 1850 and 1890 AD. Large differences in meander development and lateral migration rates were found between different river reaches. In particular in areas where non-cohesive aeolian sands formed the channel banks, large meanders formed and lateral migration reached rates up to 3 m yr⁻¹. In this research we will study two of the large meanders, named Prathoek and Junnerkoeland (Fig. 1), where Quik and Wallinga (submitted) reconstructed the scroll bar development using OSL dating in combination with historical maps.

Here we take advantage of the preservation of a palaeochannel predating the meandering phase (here referred to as "Palaeochannel X"), preserved in the Junnerkoeland as a sharp bend (Fig. 1(c)). Maas (1995) interpreted Palaeochannel X to be connected to the first swale of the scroll bar deposit before Palaeochannel X was cut off (Fig. 1(c)). Palaeochannel X was likely abandoned shortly before the scroll bar formation, because large differences in dimensions exist between Palaeochannel X and the meander bend, but the well-preserved nature suggests that Palaeochannel X is relatively young. We assume Palaeochannel X to date from the same period as Palaeochannel Q reported by Huisink (2000, p.123), i.e. prior to the meandering phase (Fig. 2), because Palaeochannel Q has similar dimensions and was already cut off on the historical map of 1720 AD. The small dimensions of both palaeochannels would suggest that the river had comparatively less energy, and may have been relatively laterally stable prior to the meandering phase.



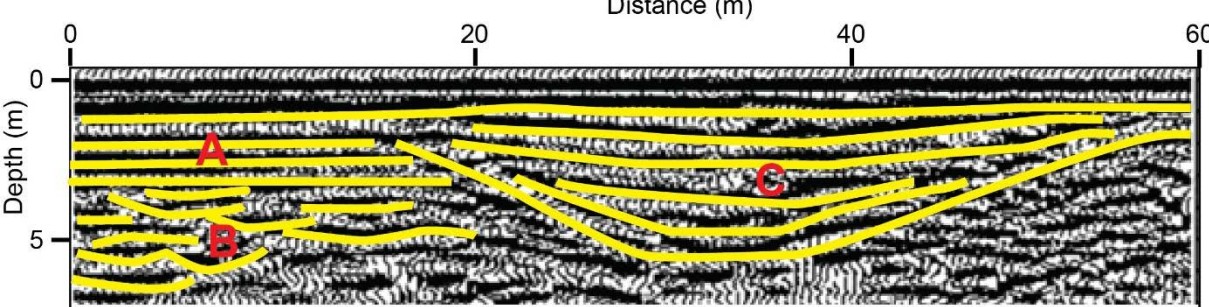

**Figure 2: Interpretation of subsurface strata from GPR data collected near Hardenberg 13 km upstream of Junnerkoeland (see location in Fig. 1(b)), modified after Huisink (2000). Horizontal strata of coversand deposits (A) on top of the channel deposits of an interpreted braiding system (B). A relatively small, symmetrical palaeochannel is present (C) within the Late-Glacial deposits, hereafter referred to as "Palaeochannel Q". Figure adapted after Huisink (2000).**

## 3. Methods

### 3.1 Approach

The first aim was to identify the possible channel pattern change. Therefore, we inferred the genesis from lithological transects in both study areas. In addition, we dated the cut-off of Palaeochannel X and investigated the lateral stability of the phase represented by the palaeochannel. The second aim was to identify the potential causes of the channel pattern change, if real. To identify whether the potential specific stream power increased, we reconstructed the bankfull discharge ($Q_{bf}$) for both Palaeochannel X and Q, and the two meanders, using the scroll bar deposition as a geological archive of the former channel dimension. The reconstructed $Q_{bf}$ is the discharge that just fills the channel before spilling on the floodplain, or the discharge at minimum width-depth ratio (Williams, 1978), and is commonly considered an approximation of the channel-forming discharge with a recurrence interval of 1 to 2 years (Wolman and Miller, 1960; Dury, 1973). The $Q_{bf}$ can be used to calculate the potential specific stream power and potential sediment transport, and hence to investigate whether changes thereof may explain the channel pattern change. A disproportionally higher scroll bar formation rate compared to the sediment transport may point at extra sediment input, which may explain the meander initiation (Ferguson, 1987; Nanson and Croke, 1992). To determine whether possible changes in discharge and channel dimensions could have resulted in channel pattern change, the potential for meandering was calculated through time using the stability diagram of Kleinhans and Van den Berg (2011) and the bar regime applying relationships of Struiksma et al. (1985), which will be further elaborated below.

### 3.2 Lithological description

Corings were performed in a transect perpendicular to the scroll bars of both meander bends (Fig. 1(c)-(d)). An additional transect was cored perpendicular to Palaeochannel X (Fig. 1(e)). A gouge auger (Ø: 3 cm), an Edelman auger and a Van der



Staay suction corer (Van de Meene et al., 1979) were used when the deposit consisted of peat, unsaturated sand or saturated sand, respectively. In total, 68 corings were performed to a maximum depth of 7.3 m (i.e. the full length of the employed suction corer with extensions). The surface elevation of each coring site was either determined using a GPS combined with a DEM (Van Heerd and Van't Zand, 1999), or with a Global Navigation Satellite System (GNSS) device. A standard method was used to describe the sediment cores in 10-cm-thick intervals, using the Dutch texture classification scheme, which approximately matches the USDA terminology (De Bakker and Schelling, 1966; Berendsen and Stouthamer, 2001). The median sediment grain size ($D_{50}$) of non-organic, sandy samples was visually checked in the field by comparison with a sand ruler. In addition, the plant macro-remains, any visible bedding and colour were described. The percentage of gravel (>2 mm) was estimated using field sieves. The lithogenesis was inferred from the lithological properties, facies geometries and DEM topography, distinguishing fluvial, fluvioperiglacial, coversand, drift-sand and residual channel-fill deposits (Ter Wee, 1966; Huisink, 2000).

### 3.3 Ground-penetrating radar

Ground-penetrating radar (GPR) was used to reconstruct the channel dimensions of the scroll bars. GPR is a suitable tool in sandy substrate (Neal, 2004), and regularly used in scroll bar deposits (Bridge et al., 1995; Heinz and Aigner, 2003; Słowik, 2011). GPR measurements were conducted with a pulseEKKO PRO 250Hz with a SmartTow configuration. The GPR transects were placed along the centreline of the meander bends, perpendicular to the ridge and swale morphology (Fig. 1(c)-(d)). The electromagnetic-wave velocity was 0.060 m ns$^{-1}$, derived by using isolated reflector points (Van Heteren et al., 1998; Neal, 2004) and by comparing depths of recognizable layers with the coring data.

### 3.4 Grain size analysis

In total 33 samples for grain size analysis were taken from the scroll bar deposits and three samples were taken from Palaeochannel X. The samples of the scroll bar deposits were taken from each 0.5 m interval from the channel lag up to the swale surface at three locations in Junnerkoeland and two locations in Prathoek (Fig. 1(c)-(e)). The samples of Palaeochannel X were taken from three locations below the residual channel-fill, from the former river bed. Grain size samples were analysed in a laboratory with a LS230 Laser Particle Sizer. This instrument has a measurement range of 0.1 to 2000 µm. Samples were sieved over a 2 mm sieve, and prepared with HCL (1 M) and $H_2O_2$ (30%) according to the laboratory prescriptions. All data were processed using a Fraunhofer.rfd optical model. Finally, the average and standard deviation were calculated for both the scroll bar deposits and Palaeochannel X, and used in the palaeodischarge calculations.

### 3.5 OSL dating

We used the scroll bar dates determined by Quik and Wallinga (submitted). We briefly describe their methods used for the OSL dating. Samples were taken to determine the burial age of the sandy scroll bars (e.g. Wallinga, 2002), using a Van der Staay suction corer (Ø 4 cm) (Wallinga and Van der Staay, 1999). The scroll bars were delineated using the same lithological

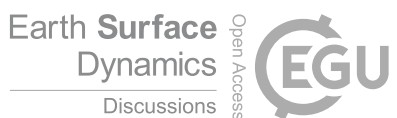



descriptions and lithogenetic interpretation as in this study. OSL samples were taken above the channel lag, and within the reduction zone to reduce uncertainty in the environmental dose rate due to water content fluctuations. The OSL age was determined at the Netherlands Centre for Luminescence dating, with equivalent doses measured on small aliquots of quartz using the SAR protocol (Murray and Wintle, 2003) and dose rates determined from activity concentrations measured using gamma-ray spectrometry. A bootstrapped version of the minimum age model (Cunningham and Wallinga, 2012) was used to

derive the best estimate of the burial dose. The thus obtained OSL ages were used as priors and combined with historical map data in a Bayesian deposition model (see Quik and Wallinga, submitted) using the OxCal software (Bronk Ramsey, 2009). The modelled age-distance relationships were used in our calculations. In this study, we took four additional samples for OSL dating on the inner and outer bank of Palaeochannel X. These samples were collected in an opaque PVC-tube (∅ 4.5 cm) mounted on a hand-auger allowing sampling without light exposure. The analysis in the laboratory followed the same

procedure as in Quik and Wallinga (submitted), apart from the final Bayesian analysis, which could not be applied to this small OSL dataset because it lacked additional constraints from historical maps.

### 3.6 $^{14}$C dating

The cut-off of Palaeochannel X was dated using radiocarbon ($^{14}$C) dating. A sample was taken in the deepest part of the palaeochannel, at the sand-peat interface. Macro-remains and leaf fragments from terrestrial species were selected from 1 cm

intervals in the laboratory using a light microscope. Samples were stored in diluted HCl. The sand content was measured for each interval to precisely determine the position of the sand-peat interface. Material with volumetric sand percentages lower than 10 to 20% was considered as peat (Bos et al., 2012). The macro-remains from the centimetre above this interface were selected for the $^{14}$C analysis providing a *terminus ante quem* date for the abandonment of the channel. The $^{14}$C age was determined by Accelerator Mass Spectrometry (AMS) at the Centre for Isotope Research (Groningen University). For

calibration, the IntCal13 curve was used in the OxCal4.2.4 software (Bronk Ramsey, 2009; Reimer et al., 2013).

### 3.7 Channel dimensions

The channel dimensions of Palaeochannel X were determined from the lithological cross-section. The residual channel-fill was delineated along the sand-peat interface. Bankfull depth ($H_{bf}$) was defined from the bottom of the palaeochannel up to the first clear knick-point on the bank, such that the width-depth ratio was minimal (Williams, 1986). The dimensions were

measured from the delineated channel, involving the bankfull width (W), cross-sectional area (A) and wetted perimeter (P). We assumed a standard deviation of 5% of the W, A and P measurements. These channel dimensions were also measured for Palaeochannel Q from the GPR profile recorded by Huisink (2000, p.123) (Fig. 2). Additional channel dimensions were calculated using Eq. 4 to 6, applying the same procedure as for the scroll bar deposits (see below).





During the meandering phase the river channel was assumed to have the channel dimensions as shown in Fig. 3. This sketch is based on Allen (1965), Leeder (1973) and Hobo (2015). The bankfull depth ($H_{bf}$) was estimated from the coring data, taken from the bottom of the channel lag up to the surface elevation in the swales (Fig. 4).  Both the bottom and surface elevation were smoothed, because small elevation differences were expected to be caused by local variation rather than real

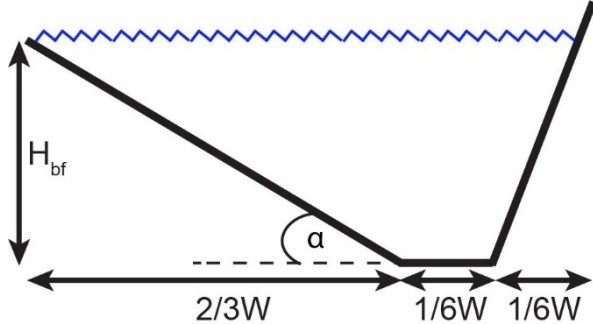

**Figure 3: Sketch of the cross-sectional flow area of a meandering channel used for the bankfull palaeodischarge calculations (Allen, 1965; Leeder, 1973; Hobo, 2015).**

changes in $H_{bf}$. We assumed a standard deviation of 5% of the $H_{bf}$ measurements, based on expert judgement. The transverse
bed slope (α) of the inner bend was determined based on the GPR transects (Fig. 5), in which lateral accretion surfaces could be distinguished. The angle was measured on the steepest parts of the identified lateral accretion surfaces. The calculations of the channel dimensions follow from Fig. 3. The bankfull width (W, m) and cross-sectional area (A, m$^2$) were determined by Eq. 1 and 2:

$$W = 1.5 \frac{H_{bf}}{tan(\alpha)} \qquad (1)$$

$$A = W H_{avg} \qquad (2)$$

where $H_{bf}$ is the bankfull depth, and $H_{avg} = \frac{7H_{bf}}{12}$ and approximates the average water depth (m). The wetted perimeter (P,
m) was calculated from the assumed channel geometry (Fig. 3) following Eq. 3:

$$P = \frac{H_{bf}}{sin(\alpha)} + \frac{W}{6} + \sqrt{(H_{bf}^2 + \frac{W}{6})^2} \qquad (3)$$

The hydraulic radius (R, m) was calculated by Eq. 4:







**Figure 4. Stratigraphic cross-sections of the study sites (for location see Fig. 1). Lithological cross-sections of Junnerkoeland (a) and Prathoek (b). Lithogenetic cross-sections of Junnerkoeland (c) and Prathoek (d) including the OSL samples by Quik and Wallinga (submitted) and OSL and 14C dating results from this study. The surface and erosive base elevation are indicated with dashed lines, resulting in the inferred water surface elevation (Hbf). (e) Zoomed-in lithogenetic cross-section of Palaeochannel X. The thick dashed line indicates the bankfull level of the palaeochannel.**



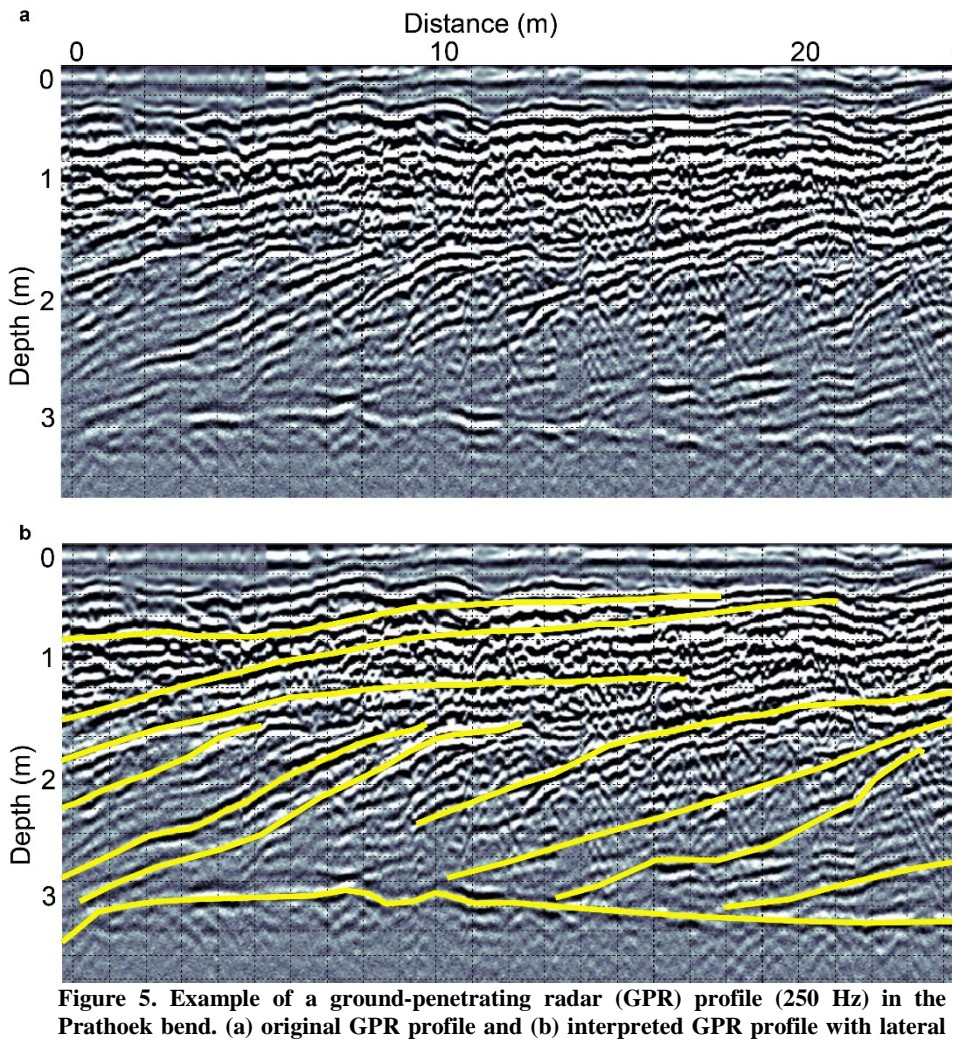

**Figure 5. Example of a ground-penetrating radar (GPR) profile (250 Hz) in the Prathoek bend. (a) original GPR profile and (b) interpreted GPR profile with lateral accretion surfaces and the channel lag, indicated by yellow lines.**

$R = \frac{A}{P}$  (4)

For each swale visible on the DEM the sinuosity (s), radius of curvature ($R_{curv}$) and scroll bar surface area ($SB_{surf}$) was measured. The former channel sinuosity was estimated by the use of the DEM, measuring the distance along the swales relative to the distance along the valley between the inflection points (Fig. 1(c)-(d)). The sinuosity of Palaeochannel X was

assumed to be similar to the start of the meandering phase, based on the assumption that Palaeochannel X was connected to the first swale in Junnerkoeland (Fig. 1(c)). The channel slope ($S_c$) was calculated from the sinuosity and valley slope ($S_v$) from Wolfert and Maas (2007) following Eq. 5:





$$S_c = S_v/s \tag{5}$$

The volumetric rate of scroll bar growth ($SB_{vol}$, m$^3$ yr$^{-1}$) was determined from scroll bar surface area ($SB_{surf}$, m$^2$ yr$^{-1}$) and thickness between each swale and interpolated time interval following Eq. 6:

$$SB_{vol} = \frac{SB_{surf}*H_{bf}*(1-\varphi)}{\Delta age} \tag{6}$$

where $\varphi$ is the porosity (here 0.35 volume fraction) (Nimmo, 2004), which was included to compare the $SB_{vol}$ with the sediment transport, and $\Delta age$ is the age difference between the scroll bars (yr) based on the datings by Quik and Wallinga

(submitted). Equation 6 was also applied to the inner bank of Palaeochannel X.

### 3.8 Palaeodischarge

The channel dimensions were used to calculate the bankfull discharge. We assumed that the bankfull discharge was similar for both Junnerkoeland and Prathoek, regarding the short distance between these river sections (Fig. 1(b)). Hence the average bankfull discharge and standard deviation was calculated, combining both meander bends. The bankfull discharge was

estimated by applying the Chézy equation, following Eq. 7:

$$Q_{bf} = CA\sqrt{RS_c} \tag{7}$$

where $Q_{bf}$ = bankfull discharge (m$^3$ s$^{-1}$), and C = Chézy coefficient (m$^{0.5}$ s$^{-1}$). From Eq. 7 the cross-sectionally averaged flow

velocity ($u_{bf}$, m s$^{-1}$) was calculated by $\frac{Q_{bf}}{A}$. The Chézy coefficient, i.e. flow resistance, is an important unknown parameter in Eq. 7. We used several methods to estimate an acceptable range for the Chézy coefficient for both the meandering and palaeochannels X and Q. The first was an empirical relation (Brownlie, 1983) following Eq. 8:

$$Q_{bf} = \left(\frac{R}{(0.3724*S_c^{-0.2542}*\sigma_s^{0.105}*D_{50})}\right)^{1.529} Wg^{0.5}D_{50}^{1.5} \tag{8}$$

where $\sigma_s$ is the sorting of the bed material grain size derived from the grain size analysis and approximated by 0.5($D_{50}/D_{16}$ + $D_{84}/D_{50}$), $D_{16}$ and $D_{84}$ are the 16$^{th}$ and 84$^{th}$ percentile sediment grain size, respectively, and g is the gravitational acceleration (9.81 m$^2$ s$^{-1}$). Equation 8 was substituted in Eq. 7 to calculate C. The second method comprised estimating the Manning roughness coefficient (n) to calculate C. We estimated the n value from the streambed characteristics such as cross-section





irregularity, channel variations, obstructions, vegetation and the degree of sinuosity, applying the procedure presented by
Jarrett (1985) and Cowan (1956). The n value was estimated at 0.0288 for both the meandering and palaeochannels X and Q,
and agrees with estimated n values for sand-bed rivers (Chow, 1959). The Chézy coefficient is related to the Manning
coefficient (Manning et al., 1890) following Eq. 9:

$$C = \frac{R^{\frac{1}{6}}}{n} \qquad\qquad\qquad (9)$$

Thirdly, we determined the median Chézy coefficient for a large dataset of 79 rivers for which sufficient data was available
(Van den Berg, 1995; Kleinhans and Van den Berg, 2011), and for a subset of 30 rivers and 20 rivers with scroll bars and
without bars, respectively. Finally, we compared the estimated Chézy coefficient with a study done on the channelized
Overijsselse Vecht (TAUW, 1992), in which the Chézy coefficient was estimated based on expert judgement.

### 3.9 Sediment transport

The sediment transport was calculated to compare with the $SB_{vol}$, which was calculated in Eq. 6. Sediment transport was
calculated in two different ways. The first method was the slightly modified Engelund and Hansen (1967) relation following
Eq. 10:

$$Q_{s,bf} = \frac{0.05u^5 Wti}{(\frac{\rho_s}{\rho}-1)^2 g^{0.5} D_{50} C^3 (1-\varphi)} \qquad\qquad\qquad (10)$$

where $Q_{s,bf}$ is the yearly sediment transport derived from the bankfull discharge (m$^3$ yr$^{-1}$), t = the number of seconds in a year,
i = the intermittency assumed to be 0.05 (Parker, 2008), $\rho_s$ = the sediment density (kg m$^{-3}$), $\rho$ = the water density (kg m$^{-3}$), $\varphi$
is the porosity assumed to be 0.35 (Nimmo, 2004). The relation of Engelund & Hansen was used, because the relation is
suitable for sand-bed rivers with relatively low flow velocities (Van den Berg & Van Gelder, 1993), and the input variables
required were available. In the second method the sediment transport was determined for each discharge magnitude and
related frequency ($Q_{s,freq}$) (Wolman and Miller, 1960) from present-day flow conditions, by assuming that the current
discharge frequency distribution also applied to the meandering phase. We used the hourly discharge data from 1995 to 2015
of the gauging station in Mariënberg (Fig. 1(b)). This gauging station is close to the study location, and has the lowest
amount of data gaps compared to the other stations. The flow duration was calculated for intervals of 10 m$^3$ s$^{-1}$, and for each
discharge interval the sediment transport was calculated using Eq. 10, excluding the intermittency factor. When the
discharge would be above bankfull, the flow would go across the floodplain. The Chézy coefficient for the floodplain was
assumed to be half the Chézy coefficient in the channel, because of the higher roughness of the floodplain compared to the
channel. We assumed that the floodplain width was 350 m for the start of the meandering phase, which was estimated from





the DEM (Fig. 1(c)), and that the width would increase proportionately with the lateral migration rate for each time step
during the meandering phase.

## 3.10 Potential specific stream power

The potential specific stream power was calculated to plot both channel pattern phases in a stability diagram. Kleinhans and
Van den Berg (2011) distinguished four different stability fields, further building on Van den Berg (1995) and Bledsoe and

Watson (2001): rivers with laterally stable channels, meandering rivers with scroll bars, meandering rivers with scroll and
chute bars as well as moderately braided rivers, and braided rivers. In this research, only the first two stability fields are
relevant. These stability fields are separated by a discriminator that represents the minimum energy needed for the channel
pattern to occur. This means that the discriminator should be interpreted as a lower threshold, rather than a hard threshold
between the channel patterns (Kleinhans and Van den Berg, 2011). The potential specific stream power was calculated by

applying the relationship presented by Kleinhans and Van den Berg (2011) following Eq. 11:

$$\boldsymbol{\omega_{pv}} = \frac{\rho g \sqrt{Q_{bf}} S_v}{\varepsilon} \tag{11}$$

where $\varepsilon = 4.7 \sqrt{s\ m^{-1}}$ for sand-bed rivers (Van den Berg, 1995). The discriminators separating laterally stable rivers from

meandering rivers with scroll bars were calculated for the measured median bed grain sizes. Multiple discriminator lines
were plotted to take into account the range in the measured bed grain sizes, applying the relationships presented by Makaske
et al. (2009) and Kleinhans and Van den Berg (2011) following Eq. 12:

$$\omega_{ia} = 90 D_{50}^{0.42} \tag{12}$$

where subscript *ia* refers to the discrimination between laterally stable and meandering channels with scroll bars.

## 3.11 Bar regime

Bar regime was predicted applying the relationships of Struiksma et al. (1985) and Kleinhans and Van den Berg (2011). Bar
regime is based on the interaction between the flow and bed sediment, and their response to disturbances. River bends can be

seen as an example of a disturbance to both the flow and bed sediment, which have different adaptation lengths over which
they return to equilibrium. This difference in response is expressed by the interaction parameter (IP, Eq. 18), which is the
ratio between the adaptation length of bed disturbance and the adaptation length of flow. The adaptation length of flow was
calculated following Eq. 13:



$$\lambda_w = \frac{C^2 H_{avg}}{2g} \tag{13}$$

and the adaptation length of a bed disturbance (m) is calculated following Eq. 14:

$$\lambda_s = \frac{H_{avg}}{\pi^2} \left(\frac{W}{H_{avg}}\right)^2 f(\theta) \tag{14}$$

where f(θ) = the magnitude of the transverse slope effect calculated following Eq. 15 (Talmon et al., 1995):

$$f(\theta) = 9 \left(\frac{D_{50}}{H_{avg}}\right)^{0.3} \sqrt{\theta} \tag{15}$$

where θ = the dimensionless shear stress calculated following Eq. 16:

$$\theta = \frac{\tau}{(\rho_s - \rho)g D_{50}} \tag{16}$$

where $\tau$ = the shear stress (Pa), calculated following Eq. 17:

$$\tau = \rho g R S_c \tag{17}$$

The interaction parameter (IP) was calculated, following Eq. 18, to determine the bar regime of rivers according to Struiksma et al. (1985), and for comparison with the theoretical thresholds of bar regime (Struiksma et al., 1985; Crosato and

Mosselman, 2009) by:

$$IP = \frac{\lambda_s}{\lambda_w} \tag{18}$$

The IP is strongly related to the width-depth ratio, and was therefore separately calculated for Junnerkoeland and Prathoek.

A low IP means that when a bar forms in response to a local perturbation, such as local curvature, the bar disappears within a short distance of the perturbation (Struiksma et al., 1985). This is called an overdamped regime and occurs in channels with a low width-depth ratio. The threshold can be calculated following Eq. 19:

$$IP \leq \frac{2}{n+1+2\sqrt{2n-2}} \tag{19}$$



where n = the degree of nonlinearity of sediment transport versus depth-averaged flow velocity. Following Crosato and Mosselman (2009) we chose n = 4, which corresponds to values for a sand-bed river. A higher IP, and hence a higher width-depth ratio, results in an underdamped regime associated with bars that also form further downstream of the perturbation. The thresholds can be calculated following Eq. 20:

$$\frac{2}{n+1+2\sqrt{2n-2}} < IP < \frac{2}{n-3} \tag{20}$$

The above described calculations (Eq. 1 to 11, and 13 to 20) were run 200 times to take into account the uncertainty of the input parameters, using Monte Carlo simulations. The uncertainty of these parameters was described above, relating to the transverse bed slope, bankfull depth, grain size ($D_{50}$), Chézy coefficient and the measured channel dimensions of palaeochannels X and Q. These parameters were used in the calculations applying a normal distribution. All results are plotted with average values from the Monte Carlo simulations, and a range of one standard deviation representing the uncertainty margin.

## 4. Results

### 4.1 Lithogenetic units

Several lithogenetic units were distinguished (Fig. 4), following similar interpretations of the sedimentary units as Huisink (2000).

**Scroll bar deposits** consist of moderately well sorted clastic sediments varying in grain size from extremely fine to coarse sand (75 – 600 µm). The colour of the unit is light brown to dark grey. Often a clear fining upward sequence is present over the depth of the scroll bar deposits, with sandy loam or loamy sand near the surface and lenses of medium fine to coarse sand containing up to 40% of gravel at the bottom (channel lag) (Fig. 4(a) - (b)). This fining upward sequence could also be recognized in the grain size analysis done for the scroll bar deposits at Junnerkoeland and Prathoek (Fig. 6). The depth-averaged grain size for both scroll bar complexes is 0.28 ± 0.05 mm. The unit can show well developed beds from several centimetres up to several decimetres in thickness. Iron oxide concretions can be found above the lower groundwater table. The unit contains fragments of plant remains, wood and shells, which are especially abundant near the channel lag. When this unit is found at the surface, it consists of a clear scroll and swale topography.





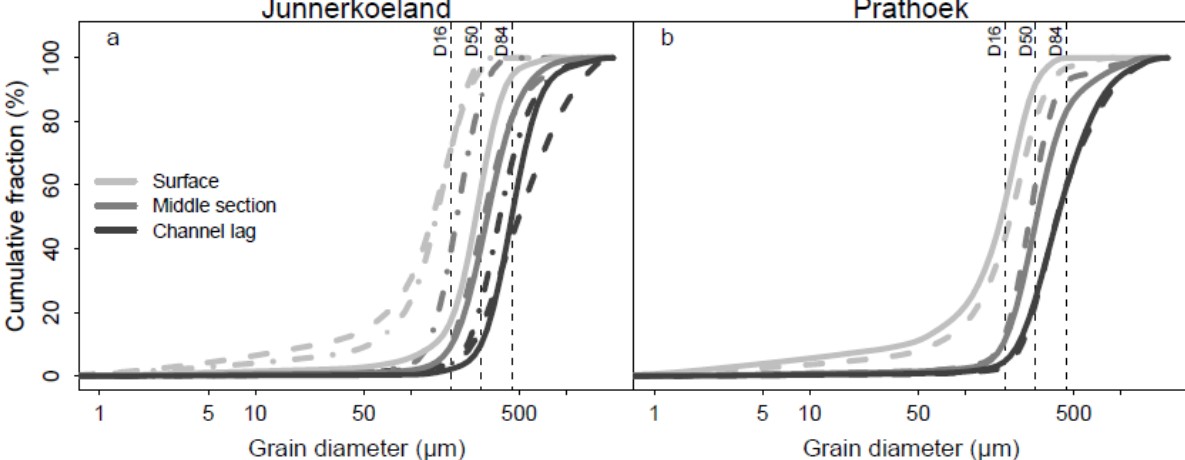

**Figure 6: Cumulative grain size distributions of the scroll bar deposits in (a) Junnerkoeland and (b) Prathoek. Three series were made for Junnerkoeland and two for Prathoek, indicated by a different line type. Each sample within a series is indicated by a different colour. The depth-averaged D16, D50 and D84 are plotted. Figure 1(c)-(d) indicates the locations of the texture samples.**

Commonly, at the base of these deposits, a sharp transition occurs to the brightly coloured substratum of fluvioperiglacial deposits below, which lack organic material. The thickness is mostly 4 to 5 metres at maximum, and the width is several hundred metres. Corings that did not reach the fluvioperiglacial deposits below the scroll bar deposits indirectly indicate the boundary between these units, because relatively resistant layers are present in the fluvioperiglacial deposits that were difficult to core into. Such a clear lower boundary was found for the southern part of the scroll bar deposits at Prathoek (Fig. 4 (b) and (d)), formed by a clay layer. This clay layer is relatively erosion-resistant, possibly limiting channel scour and thus river incision.

The GPR profiles clearly show the lateral accretion surfaces of the scroll bar deposits (see example in Fig. 5). Only where the scroll bar deposits are relatively loamy or clayey on top, the GPR results were poor (i.e. northern parts of Prathoek and Junnerkoeland). The bottom of the scroll bar deposits is mostly unrecognizable, because of a low GPR reflection at this depth. In Fig. 5 the bottom of the scroll bars is visible, because this part is located in the southern part of Prathoek where the above-mentioned clay layer was present (Fig. 4(b)), which caused a strong reflection of the GPR signal.

**Other channel deposits** were found on the inner side of the Palaeochannel X bend (Fig. 4(a) and (c)). These deposits consist of moderately sorted clastic sediments varying in grain size from fine sand to coarse sand (105 – 600 µm). The colour of the unit is light grey, light brown or white. Iron oxide concretions are abundantly present above the lower groundwater table. Small fractions of plant remains are only sporadically present near the bottom of the unit, which is slightly coarser than the upper part. Beddings are absent, as well as a clear scroll and swale topography. No lateral accretion surfaces can be observed in the GPR profile that was placed along the centreline of the Palaeochannel X bend. The thickness is similar to the scroll bar deposits, but the width is 100 m at maximum.



**Fluvioperiglacial deposits** consist of moderately sorted clastic sediments varying in grain size from extremely fine to very coarse sand (75 – 2000 µm). The unit can contain loam and loamy sand, and low percentages of gravel (0 to 20%). The colour of the unit is light grey to brown, and relatively homogeneous with depth. The unit can contain beds of loam and gravel from several centimetres up to several decimetres in thickness. Organic material is mostly absent, and only sporadically found in laminae and beds of several millimetres up to several centimetres in thickness. This unit is found below

and lateral to the scroll bar deposits, and can reach a thickness of tens of metres.

**Coversand deposits** consist of well-sorted clastic sediments varying in grain size from extremely fine to fine sand (75 – 210 µm). The unit may contain small loam fractions (<10%). The colour is light grey or light brown. Organic material is mostly absent. Soil formation, a thick palaeo-podzol, may be found in the top of this unit. This unit is located on top of the fluvioperiglacial deposits, and often near or at the surface. The unit has a maximum thickness of two metres in the study

area. This unit may be difficult to distinguish in borehole descriptions from the fluvioperiglacial deposits, when the latter has a relatively fine grain size. Because our interest is the delineation of the scroll bar deposits we combined both the fluvioperiglacial and coversand deposits into one unit.

**Table 1: OSL and [14]C dating results from Palaeochannel X. Locations are indicated in Fig. 1(c)-(d) and Fig. 4(c).**

| Sample Code | Material | Elevation | [14]C age | Palaeo-dose | Dose rate | Age | Lat, Long (RD) |
|---|---|---|---|---|---|---|---|
| | | (m +NAP) | (a BP) | (Gy) | (Gy/ka) | (ka) | |
| NCL2416194 | Fluvial sand | 1.10 | | $2.1 \pm 0.2$ | $0.81 \pm 0.03$ | $2.6 \pm 0.3$ | 229242, 505286 |
| NCL2217157 | Fluvial sand | 3.65 | | $3.3 \pm 0.2$ | $1.06 \pm 0.05$ | $3.1 \pm 0.5$ | 229249, 505254 |
| NCL2217158 | Aeolian sand | 3.99 | | $12.5 \pm 0.5$ | $1.23 \pm 0.05$ | $10.2 \pm 0.6$ | 229254, 505338 |
| NCL2217159 | Fluvial sand | 3.55 | | $3.6 \pm 0.2$ | $1.14 \pm 0.05$ | $3.2 \pm 0.2$ | 229242, 505228 |
| GrA69519 | Selected macro-fossils | 1.14 | $2300 \pm 100$ | | | $2.4 \pm 0.3$ | 229239,505298 |





**Drift-sand deposits** consist of well sorted clastic sediments varying in grain size from extremely fine to fine sand (75 – 210 µm). The colour is greyish brown. Organic material is rare. A micro-podzol may be present in the top of this unit. The unit is located at the surface, mostly on top of coversand deposits. The coversand palaeo-podzol often forms a distinct boundary between these two units. This unit is easily distinguishable from the other units, because of large topographic differences at the surface of several metres over short horizontal distances (e.g. 100 m).

**Residual channel-fill deposits** consist of (sandy) peat or peaty sand. Lenses of very fine to fine sand (105 – 210 µm), silty clay loam or clay loam may be present in the unit. The colour of the peat is dark brown, but turns black when exposed to air. Plant remains are abundantly present in the unit. Iron concretions may be present as well. This unit has a relatively low width/depth ratio (5 to 10) in the cross-section, and can have a thickness of up to four metres. In both transects (Fig. 4), peaty residual channel-fill deposits are present within the fluvioperiglacial and coversand deposits. The residual channel-fill at

Prathoek is poorly preserved and hardly recognizable at the surface.

## 4.2 Palaeo-channel X

The well-preserved Palaeochannel X is a relatively symmetrical palaeochannel (Fig. 4(e)), and very similar to Palaeochannel Q of Huisink (2000) (Fig. 2). Palaeochannel X forms a very sharp bend, which is often found in low-energy streams where lateral migration is limited (Kleinhans et al., 2009; Candel et al., 2017; Candel et al., 2018). No lateral accretion surfaces are

observed on the inside of Palaeochannel X, which can be derived from the GPR profile. The outer bank consists of Weichselian / Early Holocene deposits (Fig. 4(c)). The average grain size of the Palaeochannel X bed sediments is 0.23 ± 0.12 mm. Palaeochannel X formed by extremely slow channel displacement, shown by the OSL dates taken from the channel deposits on the inside of Palaeochannel X (Fig. 1(e), Fig. 4(c) and Table 1). A channel cut-off probably caused Palaeochannel X to become disconnected from the main river before the meandering phase started. Palaeochannel X was cut

off ca. 2.4 ± 0.3 ka, indicated by the [14]C dating (Fig. 4(c), Table 1), while inner-bend channel deposits located 50 m from the residual channel were dated at ca. 3.2 ± 0.2 ka. Hence the bend formed with a rate of ca. 6 cm yr$^{-1}$ assuming a constant channel displacement rate. The lateral migration rate of the Junnerkoeland meander bend was ca. 40 times higher (Wolfert and Maas, 2007; Quik and Wallinga, submitted).





## 4.3 Meander and channel geometry

The reconstructed transverse bed slopes do not show a trend in space (Fig. 7(a)-(b)), hence the mean and standard deviations were used in the palaeodischarge calculations. The transverse bed slope at Prathoek is higher (4.5 ± 1.0 °) than at Junnerkoeland (3.3 ± 1.3 °), but much lower than the transverse bed slope of Palaeochannel X (23.4 °) and of Palaeochannel Q (12.8 °). The age as function of distance of lateral accretion is a relatively linear relation (Fig. 7(c)-(d)), using dates from Quik and Wallinga (submitted). This relation was used for the meander and channel geometry calculations (Fig. 8). The

**Figure 7: Transverse bed slope derived from GPR cross-sections from the inner point bar to the outer bend for Junnerkoeland (left) and Prathoek (right) as well as lateral migration distance plotted against age for both bends. Panels (a) and (b) show transverse bed slope of lateral accretion surfaces measured in the GPR profile (example in Fig. 4), including the mean and standard deviation of all measurements. Panels (c) and (d) show relation between age and migration distance of the bends. Shading indicates standard deviation of the Bayesian deposition model determined by Quik and Wallinga (submitted) for the OSL and historical map dates.**





**Figure 8: Reconstructed meander and channel geometry over time, assuming the date-distance relations (see Fig. 7(c)-(d)) over the scroll bars. Panels (a) and (b) show the bankfull depth (Hbf) derived from the coring data, taken from the bottom of the channel lag to the inferred water surface (Fig. 4(c)-(d)). Panels (c) and (d) show the bankfull width for both the Junnerkoeland bend (left) and Prathoek (right) derived from the bankfull depth and reconstructed transverse bed slope (Eq. 1). The river width data from Wolfert and Maas (2007) observed on historical maps, and the bankfull river width data from Staring and Stieltjes (1848) were included for comparison. Panels (e) and (f) show the cross-sectional area derived from the bankfull width and water depth (Eq. 2). (g) Sinuosity estimated from available historical maps and DEM (Fig. 1). (h) Radius of curvature (Rcurv) derived from the DEM. (JK = Junnerkoeland, PH = Prathoek, X&Q = Palaeochannel X & Q).**





bankfull depth of palaeochannels X and Q are comparable to the bankfull depth at the start of the meandering phase at
Prathoek and Junnerkoeland (Fig. 8(a)-(b)) (3.5 to 4.2 m). The bankfull depth at Junnerkoeland decreases relatively fast at
ca. 1800 AD, because the erosive base elevation rises towards the cut-off channel (Fig. 4(c)). At Prathoek, the bankfull depth
decreases more gradual over time.

The reconstructed bankfull width of palaeochannels X and Q is much lower compared to the meandering phase (Fig. 8(c)-

(d)), resulting in a relatively small cross-sectional area of palaeochannels X and Q (Fig. 8(e)-(f)). River width observations
from previous studies were compared to the reconstructed width. These observations included observations measured from
historical maps by Wolfert and Maas (2007) and measurements of the bankfull river width over a large river section in 1848
AD by Staring and Stieltjes (1848). The river width data from Wolfert and Maas (2007) largely fall in the range of
reconstructed bankfull widths at Junnerkoeland, and show a similar decreasing trend (Fig. 8(c)-(d)). However, the historical

maps used by them may result in large uncertainties, because the water stage that these maps represent is unknown. The
measured widths by Staring and Stieltjes (1848) are in line with the predicted width at Junnerkoeland, falling within the
uncertainty range. The predicted width at Prathoek is underestimated compared to the measured widths by Wolfert and Maas
(2007) and Staring and Stieltjes (1848). This underestimation also results in an underestimated cross-sectional area (Fig.
8(f)) and consequently an underestimated bankfull discharge (Fig. 9(a)). Both at Prathoek and Junnerkoeland, the sinuosity

**Table 2: Calculated Chézy coefficients using different methods including the current Chézy coefficient of the channelized Overijsselse Vecht.**

| Method | Chézy coefficient ($\sqrt{m}s^{-1}$) | |
|---|---|---|
| | **Meander bends** | **Palaeochannel X & Q** |
| Brownlie | $48.2 \pm 0.5$ | $46.4 \pm 1.1$ |
| Manning (n = 0.0288) | $38.4 \pm 0.8$ | $39.1 \pm 1.1$ |
| Median of all rivers with scroll bars in Kleinhans et al. (2011) | 40.5 | |
| Median of all rivers without bars in Kleinhans et al. (2011) | 34.2 | |
| Median of all rivers in Kleinhans et al. (2011) | 36.2 | |
| Current channelized Vecht (Tauw, 1992) | 50 | |
| Estimation of Chézy | $43.0 \pm 2.0$ | $40.0 \pm 2.0$ |





increases during the lateral migration of the channel (Fig. 8(g)), and both bends become sharper because the radius of
curvature decreases (Fig. 8(h)).

### 4.4 Palaeodischarge and sediment transport

The Chézy coefficient was needed to calculate the $Q_{bf}$ (Eq. 7). Based on the estimated Chézy coefficients (Table 2), the
Chézy coefficient for the meandering phase should fall within the range of 38.0-48.0 $m^{0.5}$ $s^{-1}$, and for palaeochannels X and

Q within the range 34.0-46.0 $m^{0.5}$ $s^{-1}$. We used the middle of the ranges: 43.0 $m^{0.5}$ $s^{-1}$ and 40.0 $m^{0.5}$ $s^{-1}$, respectively, both with
a standard deviation of 2.0 $m^{0.5}$ $s^{-1}$ to represent the full ranges. The reconstructed $Q_{bf}$ is two to five times higher at the start of
the meandering phase (63 – 182 $m^3$ $s^{-1}$) than during the preceding laterally stable phase for palaeochannels X and Q (32 - 38
$m^3 s^{-1}$) (Fig. 9(a)). The $Q_{bf}$ declines over time, and drops relatively fast at ca. 1800 AD. The average flow velocity ($u_{bf}$) is
relatively similar for palaeochannels X and Q and the meandering phase (Fig. 9(b)) and does not change much over time.

Combining the frequency of each discharge interval with the sediment transport rate (Fig. 10(a)), results in a histogram of
the sediment transport contribution as function of discharge ($Q_{s,freq}$, Fig. 10(b)). The highest measured discharge at the
gauging station Mariënberg between 1995 and 2015 is 185.5 $m^3$ $s^{-1}$. The most frequent discharge occurring in the
channelized Overijsselse Vecht is 0 to 10 $m^3$ $s^{-1}$, with a frequency of 8.2% (Fig. 10(a)). This discharge is mainly affected by
the weirs currently present in the channelized river. Sediment transport increase is highest when discharges are low, and

decreases when the bankfull stage is reached, because additional discharge flows across the more flow-resistant floodplain.
The effective discharge ($Q_{eff}$) is 29 $m^3$ $s^{-1}$, represented by the highest sediment transport contribution (Fig. 10(a)-(b)).

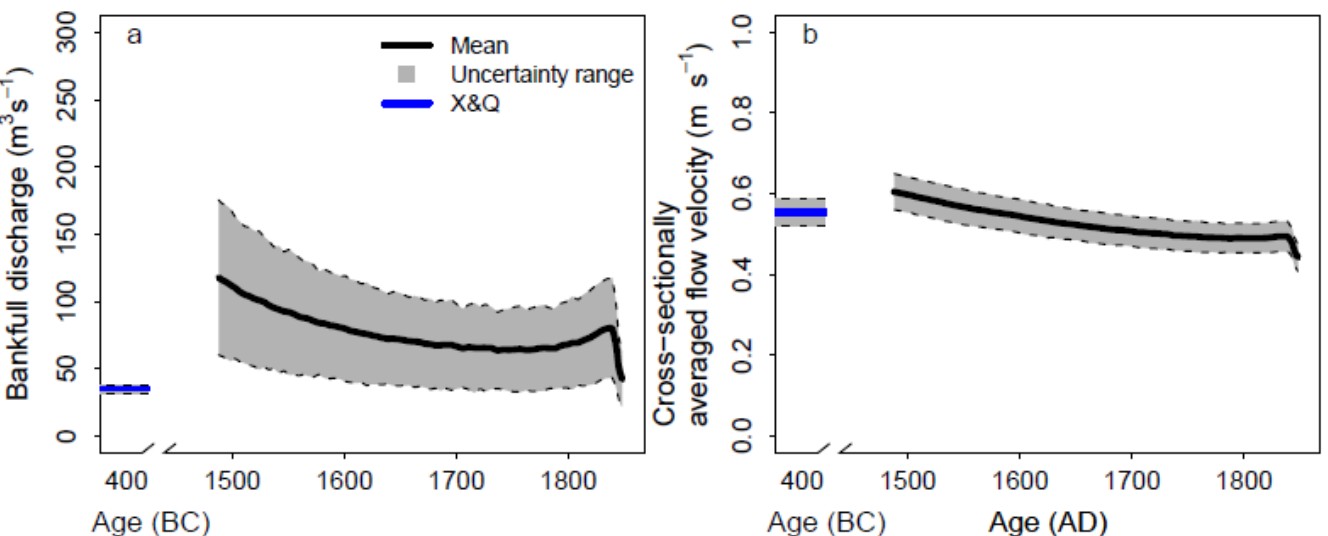

**Figure 9: Discharge and flow velocity during bankfull conditions over time, combined for Junnerkoeland and
Prathoek, derived from the channel geometry (Fig. 8) and flow resistance (Table 2). (a) Bankfull discharge (Eq. 7). (b)
Cross-sectionally averaged flow velocity. (X&Q = Palaeochannel X & Q)**





Figure 10(c) shows that both estimates of sediment transport, $Q_{s,freq}$ and $Q_{s,bf}$, were higher than the scroll bar growth in Junnerkoeland and Prathoek, suggesting that the scroll bar growth could entirely be explained by the sediment transport. Hence external sediment input was probably limited and did not contribute to the meander initiation. The $Q_{s,bf}$ of palaeochannels X and Q is much lower than for the meandering channels, explaining the large difference between the growth

rate of the channel deposits on the inner bank at Palaeochannel X (7.4 m$^3$ yr$^{-1}$) and the scroll bars of Junnerkoeland and Prathoek at the start of the meandering phase (2.5*10$^3$ m$^3$ yr$^{-1}$ and 4.6*10$^2$ m$^3$ yr$^{-1}$, respectively).

**4.5 Potential specific stream power and bar regime**

Palaeochannels X and Q seem to lack the potential to meander given their low position in Fig. 11(a), and are characterized by an overdamped regime (Fig. 11(b)). The stable character of this system is corroborated by the symmetrical channel shape,

the absence of scroll bars (Fig. 2 and 4e) and the low sediment transport (Fig. 10(c)), explaining the limited channel displacement found with the $^{14}$C and OSL datings (Table 1, Fig. 4(c)). Our data indicates that the bar regime changed from an overdamped regime into an underdamped regime (Fig. 10(c)-(d)), leading to overdeepening of the outer-bend pool and enhancement of the point bars in the inner-bend (Struiksma et al., 1985; Crosato and Mosselman, 2009; Kleinhans and Van den Berg, 2011). The higher bankfull discharge (Fig. 9(a)) explains the potential to meander (Fig. 11(a)), the high sediment

transport and the scroll bar growth (Fig. 10(c)) at the start of the meandering phase.





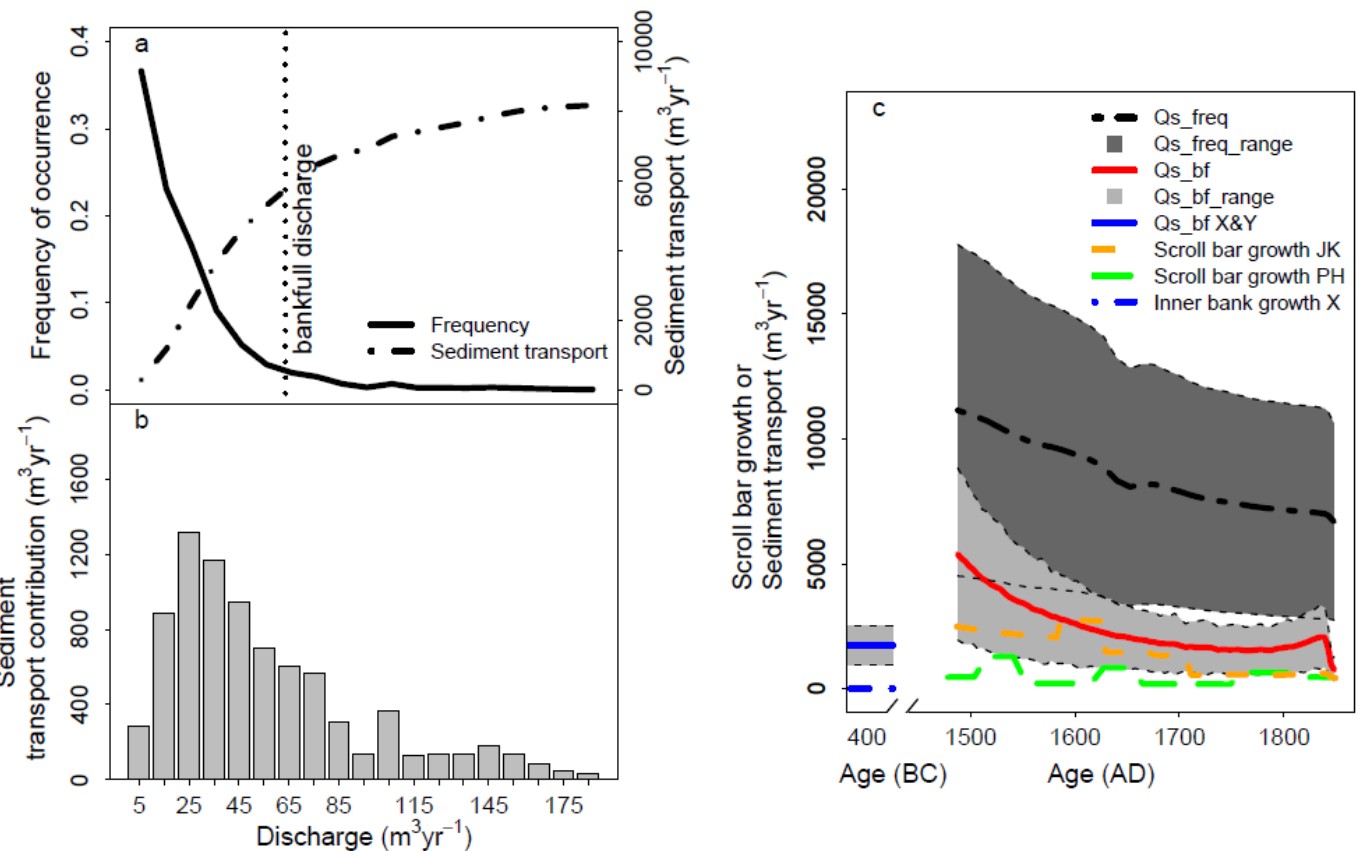

**Figure 10: Sediment transport budgets calculated from present-day flow conditions and from meander migration. (a) Discharge and sediment transport characteristics of the Overijsselse Vecht derived from hourly discharge data from 1995 to 2015 of the gauging station Mariënberg, including the frequency of each discharge class over a year, on a frequency scale from 0 to 1, and the sediment transport as function of discharge for the predicted year 1546 AD in the Junnerkoeland meander bend. (b) Histogram of the sediment transport contribution as function of discharge. (c) The sediment transport and scroll bar growth over time (JK = Junnerkoeland, PH = Prathoek, X&Q = Palaeochannel X & Q). The inner bank growth X refers to the growth rate of the channel deposits on the inner bank at Palaeochannel X, assuming a constant lateral migration rate.**

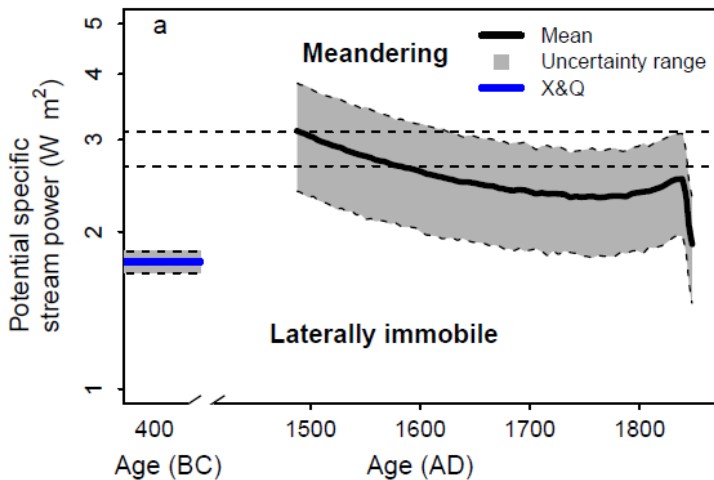

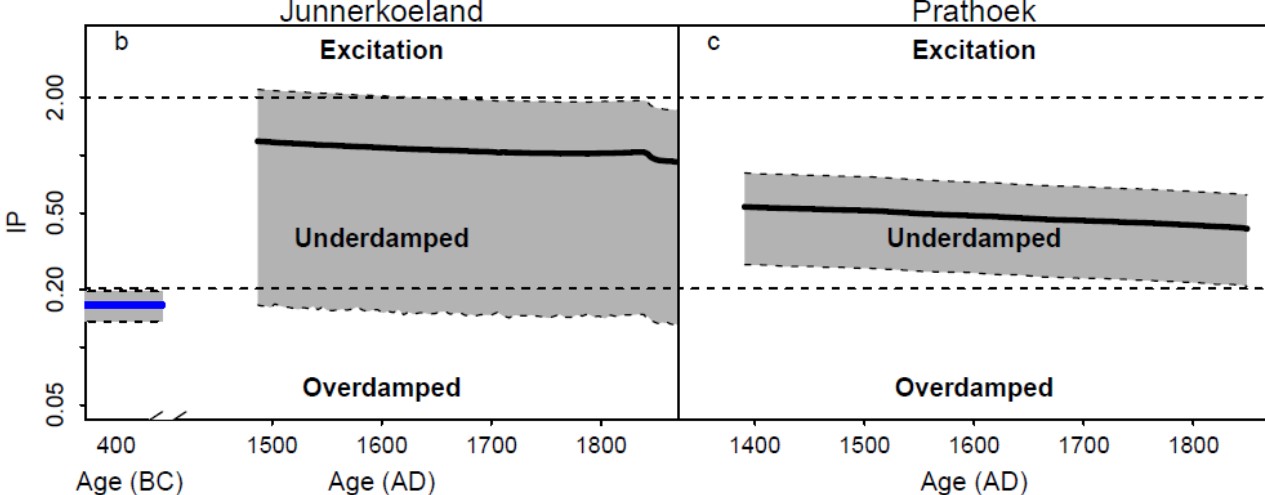

**Figure 11: The potential for meandering with time. (a) The potential specific stream power in a stability diagram (Eq. 11). Several discriminators were plotted for a range of median particle sizes of the bed sediment, which is the range of particle sizes found in the scroll bars and Palaeochannel X&Q (Fig. 6). The discriminators should be interpreted as lower thresholds rather than hard thresholds. Panels (b) and (c) show the bar regime for both Junnerkoeland and Prathoek, determined with the interaction parameter (IP) (Eq. 18), and compared to the thresholds (Eq. 19 and 20) (X&Q = Palaeochannel X & Q).**



## 5. Discussion

### 5.1 Laterally stable phase

The results indicate that the river type has changed from laterally stable to meandering. As explained previously, the preservation potential of deposits associated to the laterally stable phase is likely to be very small. Channel reaches are only preserved when they have been subject to perturbations causing them to be cut off prior to the stable-meandering transition, such as Palaeochannel X, and possibly Palaeochannel Q. In this way these reaches escaped from later lateral erosion during the meandering phase. Consequently, the lateral stability of the river is not immediately evident from these preserved channel reaches, because the perturbations led to very slow channel displacement as was found for Palaeochannel X. However, scroll bar deposits did not form, showing that the displacement was not related to meandering in which helicoidal flows cause bar formation and bank erosion at a significant rate (Seminara, 2006). Large similarities exist between the laterally stable phase reported here and the laterally stable channels in highly cohesive sediment on the intertidal mudflat, which are mostly laterally stable except from some sharp bends where bank failure and flow separation result in very limited and local channel migration (Kleinhans et al., 2009).

### 5.2 Causes of the channel pattern change

Now that we have found indications of a channel pattern change, we aim to identify the potential causes using the palaeodischarge reconstruction. It seems likely that the increasing bankfull discharge (by a factor two to five, Fig. 9(a)) caused the channel pattern change. The channel pattern change likely occurred ca. 1400 AD, because older scroll bar deposits were not found in the Overijsselse Vecht catchment (Quik and Wallinga, submitted). Scroll bar growth significantly increased as a result of higher sediment transport rates (Fig. 10(c)). The increasing bankfull discharge may reflect an increase in annual discharge, but could also be related to a more irregular discharge regime, because the bankfull discharge partly represents the higher discharges in a river (Wolman and Miller, 1960; Dury, 1973). Consequently, the discharge may have been constant over a year with low peak discharges and a relatively high base flow during the laterally stable phase, changing into a discharge regime with high peak discharges and a relatively low base flow at the start of the meandering phase. Here we discuss potential allogenic factors that may have caused such changes in discharge regime.

### 5.2.1 Little Ice Age

The Little Ice Age (14[th] to 19[th] century) (Grove, 1988) may have contributed to the channel pattern change, given the overlap in time with the meandering phase. Although geomorphological responses differ for each river during the Little Ice Age, enhanced lateral migration or incision was generally observed for most rivers in North-western Europe (Rumsby and Macklin, 1996). Studies on historical observations of nearby rivers (IJssel, Elbe, Lower Rhine and Meuse) suggested a significant higher flooding rate during the Little Ice Age compared to more recent flooding rates (Glaser and Stangl, 2003; Mudelsee et al., 2003, 2004; Glaser et al., 2010). River ice jams contributed to ca. 70% of the floods in the Rhine delta, often



5 in combination with precipitation and/or snow melt (Glaser and Stangl, 2003). These ice jams may have caused enhanced bank erosion, because ice jams can result in fast rising flow stages, whereas river ice break-ups will result in fast lowering flow stages and high peak discharges (Ettema, 2002). The water level in the bank responds fast to these changes in flow stage, hence seepage pressure will be high when the flow stage rapidly lowers. This process reduces the bank stability significantly, and may promote bank collapse of the steeper outer bend (Ettema, 2002).

10 During the Little Ice Age, the type of precipitation changed significantly, affecting the discharge regime of rivers in North-western Europe. Runoff relative to precipitation may have been higher in winter, due to reduced evapotranspiration rates and frozen soils (Rumsby and Macklin, 1996; Van Engelen et al., 2001). The snowfall/rainfall ratio was probably higher, due to lower winter temperatures in The Netherlands and Germany (Lenke, 1968; Behringer, 1999). Higher snowfall rates were also recorded for the United Kingdom (Manley, 1969), where it led to more flooding during the snowmelt period (Archer,

15 1992). In the Overijsselse Vecht catchment, snow melt probably also led to higher peak discharges. Currently, the yearly averaged precipitation over the winter months (December, January and February) is 201 mm in the study area. The largest amount of winter precipitation falls as rain, with an average air temperature of 3.4 °C for the period 1981 to 2010 (KNMI, 2010), and rapidly contributes to discharge. However, the 25-year averaged winter-temperature during the Little Ice Age was 1.2 °C, reconstructed by Van Engelen et al. (2001) for The Netherlands, suggesting that snowfall during this period was

20 much more significant. If all precipitation in winter would fall as snow in the Overijsselse Vecht catchment (3785 km$^2$), which for example would melt in springtime within two weeks, an extra peak discharge of 625 m$^3$ s$^{-1}$ would be generated when the evapotranspiration and infiltration is neglected. This snowmelt period returns more or less yearly, which matches the approximate recurrence interval of the bankfull discharge (Wolman and Miller, 1960; Dury, 1973).

### 5.2.2 Land use changes

25 An additional cause for a changing discharge regime could be land use changes in the catchment such as deforestation (Kondolf et al., 2002), which affects the discharge regime due to the direct relation with evapotranspiration (Fohrer et al., 2001). However, the most intense phase of deforestation occurred during the Iron Age and Roman period in the Overijsselse Vecht catchment (500 BC – 200 AD), as was derived from pollen records (Groenewoudt et al., 2007; Van Beek et al., 2015b). Forest was replaced by agricultural fields and open grass vegetation for grazing. Therefore, deforestation cannot be

30 the main cause for the channel pattern change discussed in this paper, because it dates from a much earlier period.

Interestingly, another major land use change occurred in the catchment at a later stage, when humans started to reclaim land in peat areas to cultivate buckwheat. This land use change started in the 12$^{th}$ and 13$^{th}$ century (Gerding, 1995; Van Beek et al., 2015a), and intensified from the 14$^{th}$ century onwards (Lenting, 1853; Borger, 1992; Casparie and Streefkerk, 1992; Van Beek et al., 2015b). Reclamation of peatlands comprised digging of channels to drain the land, and burning the top layer of

35 the peat for fertilisation. After several years the land became exhausted and abandoned, and the next tract got reclaimed (Borger, 1992). After several centuries, focus shifted from peat reclamation to exploitation, excavating large peatland areas for fuel during the 17$^{th}$ and 18$^{th}$ century (Gerding, 1995).



The cultivation and exploitation of peatlands may have had a significant impact on the discharge regime of the Overijsselse Vecht system, because approximately 27% of the Overijsselse Vecht catchment area was covered with peat around 1500 AD, of which the largest part has currently disappeared (Streefkerk and Casparie, 1987; Casparie and Streefkerk, 1992; Van der Schaaf, 1999; Vos et al., 2011). Although the reclamation was mainly limited to the margins of peatlands, the hydrological consequences were large. The margins are a natural seal of the peat bog, with a low hydraulic conductivity

compared to the remainder of the bog, ensuring peat dome growth. Destruction of these margins will result in drainage of the entire peat bog (Van der Schaaf, 1999; Baird et al., 2008). Yearly average discharges can increase by 40% in the Dutch climatological setting, due to evapotranspiration differences for reclaimed peat areas compared to undisturbed peat areas (Baden and Eggelsmann, 1964; Uhden, 1967; Streefkerk and Casparie, 1987). The discharge also becomes less well distributed over the year, with higher discharges in winter and lower discharges in summer, because water storage capacity

changes after reclamation (Baden and Eggelsmann, 1964; Uhden, 1967; Streefkerk and Casparie, 1987). Especially the volumetric storage capacity of the top peat layer changes from 80 or 90% to less than 10%, because the top peat layer gets destructed by burning and lowering of the groundwater table leading to decomposition and oxidation (Streefkerk and Casparie, 1987; Van der Schaaf, 1999).

Several studies have shown that an increased drainage network in peatlands resulted in higher discharge peaks with a fast

discharge response to precipitation (Conway and Millar, 1960; Robinson, 1985; Streefkerk and Casparie, 1987; Holden et al., 2004; Holden et al., 2006). Holden et al. (2006) found that immediately after the drainage the runoff/rainfall ratio increased, probably related to dewatering of the peatland. This response was largest immediately after peat drainage, as ditches become less efficient over time when they fill up with vegetation or sediment (Stewart and Lance, 1991; Fisher et al., 1996). Finally, canals were not only dug for peat reclamation, but also for shipping and effective generation of water power

starting in the 11[th] and 12[th] century (Driessen et al., 2000), which could have promoted the higher peak flows even more. New canals resulted in a faster runoff, but also changed the watershed delineation (Driessen et al., 2000). Consequently, peak flows as well as the total discharge likely increased due to land use changes.

### 5.3 Meandering phase

Our data strongly suggest that the changing discharge regime was the main cause for the channel pattern change in the

Overijsselse Vecht. The most likely identified causes are climate changes related to the Little Ice Age and land use changes in the catchment, in particular peat reclamation. Here we will shortly elaborate on the meandering phase, although in-depth understanding of the changes during the meandering phase is beyond the scope of this paper. Interestingly, the bankfull discharge declined over time (Fig. 9(a)), leading to decreasing sediment transport relatively to the scroll bar growth (Fig. 10(c)) and insufficient potential specific stream power for meandering after 1850 AD (Fig. 11(a)). This decline would

suggest that the forcing disappeared or diminished, and had a temporary character, which would fit with the hypothesis of the Little Ice Age that ended in the 19[th] century. However, the river was still laterally migrating until channelization in 1914 AD (Wolfert and Maas, 2007). Historical bank stability changes may have promoted the river meandering during this period.



For example, floodplains were intensively used for cattle grazing, which may have weakened the banks, enhancing meandering after 1850 AD (Trimble and Mendel, 1995; Wolfert et al., 1996; Beschta and Ripple, 2012). Also drift-sand activity was initiated by intensive land use since the Late Middle Ages (Fig. 1(c)-(d)) (Koster et al., 1993), which may have affected the bank stability. Drift-sands may also have acted as an extra sediment supply to the river, altering the river morphodynamics by enhancing the scroll bar growth rate and therefore the bank erosion rate (Ferguson, 1987; Nanson and

Croke, 1992). However, we found that the scroll bar growth can easily be explained by the reconstructed sediment transport until 1800 AD (Fig. 10(c)). Therefore, it seems unlikely that increased sediment input by drift-sands initiated the meandering, but it may have promoted meandering since 1850 AD.

## 5.4 Channel pattern changes during the Holocene

We argue that many meandering rivers with a current potential specific stream power close to the lowest empirical threshold
(Makaske et al., 2009; Kleinhans and Van den Berg, 2011) may have been laterally stable during most of the Holocene, prior to the Little Ice Age or increased human activity. In general, modified land use and water management for agriculture and urbanization purposes often caused increased high discharge peaks and flooding (Leopold, 1968; Fohrer et al., 2001) and hence increased fluvial activity. In particular, Wolman (1967) related fluvial changes such as channel widening to land use changes, and since then many studies followed (Gregory, 2006; Notebaert and Verstraeten, 2010; Downs and Gregory, 2014;
Notebaert et al., 2018). The same applies to the consequences of increased snowfall and ice jams to fluvial activity during the Little Ice Age (Rumsby and Macklin, 1996). However, collected evidence of pattern changes of laterally stable to meandering during the Holocene is limited (Lewin and Macklin, 2010), which may mean that evidence is poorly preserved and/or not interpreted. Also, sinuous laterally stable rivers may have been misinterpreted as meandering. To identify Holocene channel pattern changes, a higher resolution of subsurface data is needed than usually gathered, because
palaeochannels of laterally stable rivers are poorly preserved in the fluvial archive of meandering channel belts (Van de Lageweg et al., 2016).

The Geul river in the southern Netherlands is an example of a river where a similar channel pattern change from laterally stable to meandering may have occurred. De Moor et al. (2008) hypothesized that this river was relatively laterally stable during the Early and Middle Holocene, until the last 2000 years in which the river was actively meandering. Most of the
floodplain deposits from the laterally stable phase have not been preserved, but De Moor et al. (2008) were able to reconstruct the bankfull depth for both periods. They estimated the bankfull depth to be a factor two to three higher during the Late Middle Ages compared to the Early and Middle Holocene, caused by human and climate impact. Although they argue that their evidence is limited, our insights support the likelihood of their findings.

## 5.5 Implications for river management and restoration

A better understanding of channel pattern changes is of major importance for river restoration, which turned into a multi-billion industry aimed at improving the ecological functioning of rivers (Sear and Newson, 2003; Malakoff, 2004; Lewin and



Macklin, 2010). Meandering rivers are often preferred in river restoration because of the high ecological value of such riverine landscapes (Ward et al., 2001; Ward et al., 2002), but the necessity of sufficient stream power to induce lateral migration is often ignored (Kondolf, 2006), and knowledge on the planform evolution is often lacking (Wohl et al., 2005). River restoration will result in increasing numbers of rivers where artificial bank protection is removed and natural processes can thrive. Although most lowland rivers are relatively laterally stable after restoration (Eekhout et al., 2015), they could

change into meandering rivers when external forcings change, e.g. related to future climate or land use change (Anisimov et al., 2008). On the other hand, water retention measures in the catchment aiming at reducing flood risk or enhanced groundwater recharge, could result in a shift from active meandering to laterally stable rivers. Considering the importance of land use and climate on the river channel pattern, it is crucial to align plans for future landscape design and climate projections with river restoration goals. Therefore, predicting and understanding channel pattern changes is important to

allocate sufficient space for rivers and to protect infrastructure from fluvial erosion, or to take precautions in the catchment to mitigate the impact of climatic changes on the river discharge regime.

River restoration measures should be approached from the processes (Brierley and Fryirs, 2009; Brierley et al., 2013; Makaske and Maas, 2015), rather than from a historical reference (Dufour and Piégay, 2009; Bernhardt and Palmer, 2011). The stability diagram and bar regime theory offer an easy approach in the restoration of low-energy rivers to estimate the

potential for meandering. Our analysis shows that these empirical tools work relatively well to discriminate the laterally stable channel pattern from the meandering channel pattern (Fig. 11).

## 6. Conclusions

The channel pattern of the Overijsselse Vecht changed from a laterally stable into a meandering river during the Late-Holocene. We attribute this change to a two to five times increase in bankfull discharge, based on a palaeodischarge

reconstruction building on channel dimensions of the different phases. Consequently, the river had sufficient potential specific stream power to erode outer banks and sufficient sediment transport to build scroll bars, in contrast to the preceding laterally stable phase. The bar regime changed from an overdamped to underdamped regime, leading to overdeepening of the outer-bend pool and enhancement of the point bars in the inner-bend. Historical land use and climate change were identified as the most likely causes of the channel pattern change. The bankfull discharge increased partly as a result of the Little Ice

Age, due to increased snowfall and ice jams. Moreover, peat reclamation and exploitation has contributed to a changing discharge regime, as well as the digging of new canals for shipping and effective generation of water power. We argue that similar channel pattern changes likely occurred in many other low-energy rivers during the Late Holocene, but these are difficult to identify due to poor preservation of channel deposits associated with laterally stable river phases. Considering the importance of land use and climate on the river channel pattern, it is crucial to align plans for future landscape design and

climate projections with river restoration goals.



**Acknowledgements**

This research is part of the research programme RiverCare, supported by the Netherlands Organization for Scientific Research (NWO) and the Dutch Foundation of Applied Water Research (STOWA), and is partly funded by the Ministry of Economic Affairs under grant number P12-14 (Perspective Programme). M.G. Kleinhans was also supported by the NWO (grant Vici 016.140.316/13710). The authors would like to thank the following persons for their help with the different

methods used in this research: Joep Storms, Gerard Heuvelink, Marijn van der Meij, Wobbe Schuurmans, Alice Versendaal, Aldo Bergsma, Marjolein Gouw-Bouman and UU-students Karianne van der Werf, Sjoukje de Lange, Jip Zinsmeister, and Pascal Born. We would also like to thank Jan Roos, Staatsbosbeheer and waterboard Vechtstromen for the access to and the insight knowledge of the field sites.

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
