# Peer review of "Late Holocene channel pattern change from laterally stable to meandering caused by climate and land use changes"

_Earth Surface Dynamics, 2018_

## Referee Comment (RC1) · Anonymous Referee #1 · 11 Jun 2018

I read a manuscript in a very well defined shape. The language, organisation, amount of references and overall quality is high and if the figures can be polished/optimised, the technical part will be of very good quality. I consider the topic of the study timely, relevant and well placed in the scope of the journal. The methods appear to be mostly adequate and thoroughly described. I especially welcome the general attempt to account for the uncertainty inherent to several of the inferred parameters, although there are further uncertainties that should be added to reach a more comprehensive capture of the total model uncertainty.

Technically, I find the use of dotted and dashed lines in many of the figures disturbing.

They make it sometimes very difficult to actually see the data that is to be visualised. For example in fig. 10C the dashed lines obscure the course of the data almost completely. Please think of reworking most of the dotted and dashed lines. In many cases they are not needed to make a distinction in the plots.

I strongly encourage the author(s) to provide along with the study also the code and data they used to generate their results. This would make it possible to reproduce their work and also increase the impact of the study. I have not doubt about the validity, rigour and correctness of the material but without seeing it I can hardly judge these points. Beyond that, readers of the paper will be happy to already have a starting point to proceed with if the code and data were presented along with the article.

I had the impression that there are some sections that are too inflated with information and detail, much more than what is actually needed to support the statements they are about to make. For example, the study area section, especially the restoration part, is interesting to read but very detailed, as well. Please consider restricting the content to what you essentially need to support your methods and the subsequent discussion/interpretations.

Likewise, there are results reported in great detail that are not used to a reasonable extent, any more. The classic example for this is section 4.1. Such details may become part of supplementary materials but unless you need this for the discussion, it is not needed in the scope of the manuscript.

The abstract is mostly clear and gives a good overview of the topic and the main findings and their interpretations. It should however shed some more light onto the most detailed part of the study: the development and application of the calculus to describe hydrologic parameters and channel metrics.

The introduction is well organised and follows a consistent flow of context. The references might imply that it is almost exclusively Dutch scientists that have worked on that topic. If that is the case, fine. If not, it might be good to also present adequate

references from other regions. But this is just a suggestion that may help improving the manuscript.

The scope of the study as expressed at the end of the introduction is not a good match with what I read later on. The actual study goes way beyond the short summary of "detecting channel pattern change" and "identifying causes". Please give more details about the approach, as well. The field and especially the numeric work is a considerable and innovative part of your work and should be reflected by the scope definition.

Concerning the second part of the scope ("identifying causes"), this part is not ideally resolved, neither by your data nor by the discussion. In the latter part, you mainly cite other people's work and make a proposition that the Little Ice Age meteorological conditions and/or land use changes have had an influence on the observed/modelled results. But you do not and cannot easily go beyond this general statement. So maybe this part of the scope should not be a central goal?

The study area description is fine, though in parts a bit too long. Please see detailed comments.

The field methods description is in most cases conclusive and well understandable. See detailed comments below for some adjustments.

The calculus description is less consistent. I acknowledge the idea of accounting for parameter uncertainty. But this must be done comprehensively and with justification. For several parameters there are either no uncertainties given or they appear out of the blue. See details below.

The order of the equations does not match the order in which the text refers to them. So either re-order the equations or tweak the text to match the equations.

The Chezy coefficient was assumed/estimated by several approaches. This is fine but in the discussion the average of all these approaches was used as the most likely value. I do not see a justification for this attempt. Are all these approaches equally likely or

equally valid? If not, how and why was the final average coefficient value estimated?

Overall, sections 3.7 – 3.11 introduce a large set of assumptions and equations/models. These are not well reflected in the introduction and scope of the manuscript. So, do you really need all these models to make your points and interpretations, or the other way around, are your research questions adequately addressed in the beginning to prompt such a large set of concepts and models?

The set of parameter values were sampled and computed 200 times in a Monte Carlo approach. Are you sure that 200 MCMC runs are enough to cover the effects of variability adequately enough? From my experience with models that contain way less parameters I always needed much longer Markov chains to reach stable uncertainty estimates. Can you show that 200 is ok? Or have a test of convergence with number of model runs?

The results are mostly well presented. However, section 4.1 gives a very detailed picture of the lithology that is not used later to an extent that would justify this detail. I suggest to move this section to the supplementary materials to keep the story of the manuscript tight enough to be followed easily. Alternatively, make better use of the details in the discussion.

The discussion sections should be reorganised to be more logical. I suggest to focus on time and not necessarily flow of context. You can/should start with the "laterally stable phase", then "channel pattern change", then "meandering phase", then "channel pattern reorganisation". This would keep the chain of information much more concise. You can implement sections like 5.2 into this system. I would also suggest to shorten section 5.5 considerably and have it as a conclusion theme. See details below.

Sections 5.2.1 and 5.2.2 are very detailed but mainly bring together findings from other studies, focusing on potential impacts of climate change and land use change. Please shorten and condense it to what you actually need to support your findings. It would be much more appropriate to have these two sections organised together with section 5.2

(causes of channel pattern change) but also to make more links to your actual results.

Actually, it is not really possible to disentangle the effects of "Little Ice Age weather" and "land use change" from your data situation. It can be either or both that may have drive your system of channel pattern change. Please mention this issue. It is no problem to have the effect of both.

P 1, l 13-14, "changes in climate or land cover". There are certainly more that just these two drivers that can lead to changes in a regime. Consider changing to " changes in, for example, climate or land cover".

P1, l 17, "proven" is not a good term in the scientific approach. Consider replacing by "constrained".

P2, l 20, consider changing "are documented of channel pattern changes" to "of channel pattern changes are documented".

P6, l 6-7, hard to understand the value assignments. Consider rewriting to "with an average annular discharge Qm of 22.8 m$^3$/s and a mean annual flood discharge Qmaf or 160 m$^3$/s".

P6, l 15-22, too detailed. Consider shortening significantly to an extent that matches the scope of the study.

P6, l 31-35. Actually all you can say is that the cutoff happened before 1720 AD. There is no information that supports the statements like "shortly before" or "date from the same period". Consider rewriting to stay with the available constraints.

P7, l 23, check overall the journal's definition of figure reference format rules, i.e., if "(Fig. 1(c)-(d))" is the right way.

P8, l 6. The use of "respectively" makes it very hard in this sentence to identify the cases in which you used which device. Please rewrite like "In case we we used this device. In case B we use that device".

P8, l 12, did I read this correctly, that you sieved material from a 3 cm wide auger/corer to estimate the gravel content? Is this a representative sample size, or in other words, over which depth interval did you have to average to get sufficient material for sieving?

P8, l 28, Add manufacturer info to grain size device (Beckman Coulter, Malvern, Horiba, etc.) to make clear which device you used.

P8, l 29. Check units. Is it 2000 m or $\mu$m?

P8, l 30. Why was the Fraunhofer model valid? Was it "just" sandy material with minimum clay content? If not, the Mie model might be more appropriate.

P8, l 33-34. Consider rewriting to simplify. E.g., "We used the scroll bar OSL burial ages determined by Quik and Wallinga. For details on the method see this reference."

P9, l 16, what is the consequence of the different age determination procedure for the palaeo channels? Are the Baysian constrained ages comparable to unconstrained ages? Are just the errors larger?

P9, l 18, how were the radio carbon samples taken? From a corer or a pit?

P9, l 21, add HCl concentration

P9, l 31, Why did you assume a standard deviation of 5 %? Why this value? Does this come from the uncertainty arising from the GPR results? It should at least be justified somehow. Otherwise I could ask, why was it not assumed to be 0.5 % or 50 %?

P10, l 9, same as above, why the 5 %? Can you say something beyond "expert judgement"? It would considerably improve the impact and value of the study and since there are quite large uncertainty ranges in some of your results these input uncertainties may be crucial to evaluate the results. You can for example also think of sensitivity analysis. What would happen if you set the standard deviation to 1 % or 15 %?

P14, l 13. Is there any uncertainty available for the porosity value? Can you estimate a plausible value?

P14, l 14. Is there any uncertainty available for the age differences? Yes there is. So this should be included in the MCMC approach.

P14, l 24-25, the sentence does not fit very well, here. Consider shifting it to a more appropriate place where it does not cause a break in context.

P15, l 6, Is there any uncertainty available for n? Can you justify why you chose 0.028 for this parameter?

P15, l 12, Which type of rivers were these 79? sand bed? low land? Some detail is needed to understand the validity of averaging over this number of rivers.

P15, l 15, Who was the expert that suggested the value of the Chezy coefficient?

P15, l 24-25, give uncertainty estimate for intermittency and porosity parameters. Or say there is no uncertainty.

P15, l 27, consider new paragraph between "available" and "In the second".

P16, l 8-14. This is vital information about the stability diagram. Please deliver this earlier to the reader, e.g., when you first mention this diagram type. What is meant by "interpreted as a lower threshold, rather than a hard threshold"?

P21, l 12, define or quantify the term "very similar", you have the data to do so.

P21, l 17, define or quantify the term "extremely slow", you have the data to do so. Also, you can make use of the uncertainty information.

P22, l 8, provide uncertainty information for slope of X.

P22, l 11, provide uncertainty information for slope of Q.

P22, l 11, what mean "relatively linear"? You should test and quantify. Actually I could also interpret a piecewise linear model with a break around 1850.

P25, l 8-10, why did you choose the "middle of the ranges" and what is the "middle"? See above, why should the full range of estimated values for C be equally valid or

likely? If they were equal, why would you make a distinction between "all rivers", "rivers without bars" and so on? Why did you use a standard deviation of 2 units? Please justify these apparently arbitrary assumptions. It is fine to include uncertainties, but their foundations must be reasonable.

P25, l 12, the values 32 and 38 are really really hard to map out on figure 9 a. And anyway does not everything in this figure drown in the uncertainty polygon? Please discuss your values with respect to the large uncertainty range.

P26, l 7, "was probably limited"... not necessarily. You simply cannot resolve this statement with your data. Just that the phenomenon could be explained with option A (sediment transport is higher than bar growth) does not mean that option B (external sediment input) is not also contributing. Or would these two options be mutually exclusive?

P28, l 7-14, this part contains very limited information but instead many repetitions of already discussed material.

P28, l 19-28, there is a lot of general information and unknown statements in this section. Please make a better connection to the results section. You have a lot of quantitative results, so please use them to support the statements made, here.

P30, l 31-32, if this is no tin the scope of the paper, then why referring to this topic?

P31, l 11-12, I do not think it is actually possible to resolve whether or not increased sediment input played a role, so I would not mention this, here. See comment somewhere above.

P31, l 14-33, very broad and general. The main point I read from this paragraph is that we need more detailed field studies to pursue the question. Try to make more out of this material. It would be a valid goal to investigate if the one case you found in your study is an "outlier" or the "regular case". Anyhow, the paragraph in its current shape does not present/discuss your results. You have to make a story out of it or leave it.

Likewise, the second paragraph comes a bit out of the blue. How does the Geul river come into play and why does it come into play, here? This section needs more context or should be skipped. Currently, it does not really match the section header.

P32, l 5-15, this part is also very broad/general and arm waving. Consider shortening significantly and link it much better to your concrete findings, i.e., what your case study can contribute to this overall picture. Overall, I suggest to shorten this part and have it rather a conclusion item than part of the discussion.

P32, l 29, change "discharge increased" to "discharge potential increased".

P32, l 30, change "exploitation has contributed" to "exploitation has probably contributed".

Table 1, It would be better to have the radio carbon and OSL ages at the same scale. This concerns both, years versus kilo years and AD years versus absolute years. At the moment things are hard to bring together.

Figure 1, replace dashed lines in panel b and f. Also, consider using solid lines to illustrate the zoom from panel to panel. Add similar "zoom lines" also from b to c and b to d. Provide a solid or at least partly transparent background to legends. The legend contents are really hard to see. Add legend frame in panel b.

Figure 2, image quality is not good. Either this is due to the manuscript stage compression or other. It would be essential to add a higher resolved image of the GPR output. Also, the thick yellow lines are masking the raw data too much. Figure 5 does a much better job by showing both, raw and interpreted options. Alternatively, think of using thinner and semi-transparent lines. In figure caption there is repetition with "modified after Huisink" and "adapted after Huisink".

Figure 4, Please decrease the size of the drawings and have all of them on one page. The context density of the drawings is not too high, you can scale them smaller without loosing much of the content. Of course the axes labels and plot drawing texts must be

rescaled to an appropriate font size. But currently, there is a mismatch in the size of the figures with respect to what they tell.

Figure 7, what do the errors want to tell in panels c and d? Overall, the resolution of the images are not really great Consider saving such plots are EPS vector data.

Figure 8, figure quality/resolution is bad. Please avoid the dashed and dotted lines (e.g., panel g), they make it hard to see the data clean. Shift legend from panel a to panel c and d.

Figure 10, dashed lines make it hard to see any trends

Figure 11, why is Prathoek missing in above panel?

---

## Referee Comment (RC2) · Anonymous Referee #2 · 12 Jun 2018

The manuscript "Late Holocene channel pattern change from laterally stable to meandering caused by climate and land use changes" aims to identify river channel pattern changes using sedimentary and geochronological data and to identify causes for these changes. The manuscript is well written, the topic is relevant and in the scope of the journal, and the concepts and ideas are sufficiently novel. The methods are consistent and well described. There are some minor to moderate shortcomings, listed below. When these shortcomings are resolved, I consider this manuscript as a valuable contribution to Esurf.

- Some sections are written too extensively, and not all information is needed to answer

the research questions. For instance, the details on river restoration in section 2 are not needed and can be limited to a minimum. Also section 5.4 and 5.5 can be shortened.

- Section 2 (study area): P6, L29-37: A lot of assumption are made in this part. I suggest to move this part to section 4.2 (results). And then in section 4.2, you have to provide all available arguments to state that channel X is predating the meandering phase. Show data to support your statements (eg show the GPR profile). You have to provide good arguments to state that channel X is from a laterally stable phase, since this is an important point for the rest of the story.

- Section 3.1 is not needed to my opinion. Aims are already explained in section 1 (Introduction); methods will be described in detail in the next paragraphs (3.2 and next sections).

- P9, line 29: How did you define the knick-point on the bank? What will be the effect on bankfull depth and discharge when using a different knick-point on the bank? You can try a sensitivity analysis to check the effect of the definition of the bankfull depth.

- P9, line 31: Why a standard deviation of 5%? Which arguments do you have? This is an important point, since large parts of your interpretations are based on this standard deviation. If you assume a standard deviation of 10 or 20%, it is possible that your differences explained in figure 8 are not so clear anymore. Can you provide a consistent method to define the standard deviation? Also here, you can try a sensitivity analysis to check the effect of the standard deviation.

- Same question for P10, line 9.

- Section 4.1: You can summarize this section in a table showing the most important characteristics of the different lithogenetic units. The table can then be followed by a short paragraph on defining the scroll bars and scroll bar dimensions.

- Section 4.4: L11: Use statistical tests to check if the reconstructed discharge differs significantly. Given the uncertainty range it is possible that you can not reject the null

hypothesis (Q does not differ). The same for L13: 'Q drops relatively fast at 1800 AD': Given the uncertainties, it is possible that Q is not significantly different. Use statistical tests to support your statements.

- P 29, L20: It is also likely that the discharge does not differ significantly, given the uncertainties. See my previous comment.

- Section 5.2: This section mainly brings together results of previous studies and it is not based on new data. So this section should be shortened and should link better to your own data and findings. Try to better link quantitative data on climate change and land use changes with your findings.

- Section 5.2.2: Is there an observed increasing in urbanization in your catchment? Urbanization can cause higher peak discharge, which have been described in catchments in The Netherlands.

- P31, L6 and L11: 27% of the catchment was covered with peat + yearly average discharges can increase by  $40\% \Rightarrow$  ca. 11% increase in average discharge for the entire catchment. How does this compare to your reconstructed increase in discharge?

- P 31, L 29-31: "Our data strongly suggest": not correct. As you stated in section 5.2 it is likely that the increasing discharge caused the change; you have some good suggestions but no hard evidence. "The most likely identified causes": actually these are the only factors checked. You did not checked other contributing factors.

- Figure 4: Indicate the location of the datings on Figure 4e.

- Figure 10c: this figure is not entirely clear. The dashed lines do not help. Try to simplify this graph to make it more clear.

- References: For some references, correct volume, issue and pages are missing: P36, L5-6; P36, L24-26; P36, L56-57; P37, L40-41 (I may have missed more).

2018.

---

## Referee Comment (RC3) · P. Houben (Referee) · 13 Jun 2018

P. Houben (Referee)

p.houben@luc.leidenuniv.nl

The paper on "Late Holocene channel pattern change [. . .]" by Candel et al. reports on the use of floodplain stratagr. records and chronologies to conduct a quantitative assessment of (paleo)hydrological channel planform change over the past 600 years in the NE NL. At a general level the manuscript is organised, the introductory sections provide background to the research, and give sufficient detail of the used methodology. The methodological approach and the subsequent evaluation of obtained results are based on a strong research effort, and the discussion puts the work in the context of previous work and addresses potential implications. All of which fits the journal's scope.

All in all, the ms represented a valuable contribution to ESurfD, however, in its current form it requires restructuring and (partially) rewriting at the paragraph level. Regarding given standards a number of statements are misplaced. For example, the Results section includes discussions of the findings, which is why the actual Discussion mostly reverts to a sometimes narrative analysis. The weakest sections, thus, are the Discussion and the Conclusions wherein some thoughts brought up and connections that are sought to be made should be reconsidered with respect to whether they actually add to the paper's significance. In consequence, the abstract should be rewritten because it is not reflecting the actual paper content (and the balance of the featured aspects), and the highlighted findings are not supported by the employed methodology. At places abundant in-text citations in the Introduction can be perceived as a bit too excessive.

Key: rm - remove; rw - rewrite/reword; p1: Title: Actually, the paper does not include hard information that allows for pointing to the actual causes of the described channel change. In the paper, a number of (truly) possible and plausible causes are mentioned but no conclusive evidence can be shown that helped to causally link channel change to either or both of the drivers. Why not highlighting the strength of the paper, the application of quantitative palaeohydrological approaches to answer the actual research question?

- The Abstract . . . "related to changes in climate and/or land"

15-18 - Results are reported before the actual scope of the paper is given. And the approach is only explained later on. Rearrange to present a logical flow.

- Actually, no potential causes have been investigated. This is misleading information. Only other people's work is cited in the Discussion when attempting to explain what possible causes have been around. The nature of that discussion, nevertheless, remains speculative.

-29 - 'reflecting relative ...' this statement should be rephrased because it it ambiguous, and overall not intelligible when only reading the abstract.

- The last sentence is not specific to the paper content, rather will appear like a motherhood statement the the journal's audience. Remove and replace it by strong statements that stress the significance of the own findings. The reason for the weak end of the Abstract, my guess, is the underdeveloped Conclusions section (see below).

- 'Several ...' Sentence can be deleted.

p2 5 - In a braided river system, isn't the temporary presence of laterally stable/migrating channels (runnels) just a matter of stage at a time?

- ' variables like potential ...'

- rm: ', which is ... slope'

- '2011), bank erodibility (...), cohesiveness (...), and by vegetation (...).'

9-10 - rm: 'which is ... (Turowski, ...)'; 'that can increase ... '

vs 11 - Statements contradict each other

- rm: gradually

13-19 - shorten para

- rw: 'the exception is formed by human intervention'

23-34- This para does not fit in here. The surrounding text provides background information that should translate into the 'gap' and clearly formulated research goals, however, this para explains processes of channel change. Could be moved together with p2 10-18 to line 18 on p3.

- Excessive citing ... Can the information be organised into a table?

p3 11 - It feels as if already here the paper's research question is addressed, but the authors then return to reviewing literature.

- Shouldn't the information be part of the first para on p3?

- '.. stable channels poorly preserve except for ...' rm all the rest between 21 and 25

- ' Huisink, 2000) while the meandering pattern has remained throughout ...'

- rm: 'However'

p6 14-16 - This needs to be moved to the Intro. There, it was already used to justify the research effort. In general, most of the content of p6 should be part f the Intro because it is the background against which the present investigation can be justified. (I.e., it's potential value to inform restoration projects.) This is even more important as this point is picked up in the discussion s one of the more significant implications ...

- In far can could the used features by local peculiarities due to their peculiar morphological context?

p7 7 - First sentences should not lead the Methods sections. Stating the paper goals belongs to the Intro.

7-21 - The whole para is a mix of review (again) and methods description. Needs to be rectified. Fig. 2, A - B - C designation is hardly readable.

p8 6 - r: ' (i.e. the full ..)'

- Estimating a statistical parameter for which others apply stacks of sieves by just visual(?!) means? That might work depending on what the information is used for. For me this is a point for of major concern. Actually, the D50 value is key to the calculations performed employing eq. 8, 10, 12, 15, and 16.

While in general the methodology also accounts for ranges or error, I am not convinced that the 5% uncertainty is fair for this error-prone guesstimate. How good (=reliable,

=reproducible!) can the far-reaching conclusions drawn be? (E.g., see fig. 10).

17-19 - rm: 'GPR . . . 2011).'

- replace: over -> with

- What sort of laboratory prescriptions (= 'instructions')? Sounds like voodoo science, doesn't it?

33- rm: the

- rm: 2nd sentence

- 'The scroll bars' . . . can be removed, or reword, or ..

p9 18 - rm: first sentence

21-22 rm: whole sentence, it's just nomeclature

- Why 5%? Can you justify this? Still a rather optimistic estimate.

p10 Insert space between Fig. 3 and the text. The figure even may be left out.

p11, 12 Nice figures. However, would it work for people who printed it in B/W?

p14 29-32 - How was D16, D84 determined? Also visually? From the waterlogged sands that spread to either side when the sample material is pushed out of the Vander-Staay tube? I think this is a soft point of the methodology, in particulary with respect to the heavy mathwork that follows to nail physical, hydraulic parameters of in-channel water and sediment flow.

p15 15 - State what was actually used here. Rather an issue of the methodology than a result.

- New para.

p16 28-33 - rm: 2nd sentence p18 All in all, the whole Methods section could be more concise, focused. It would be worth to focus on the most important aspects and move the remainder to the Appendix.

p19 6-11 - Reword.

- rm: 'Such a clear . . .Prathoek'

- rm: last sentence

- rw: abundant above

Whole section 4.1.: Commonly, the ordering of geol. units is from old to young.

p21 22-28 - 'Palaeochannel . . .' All this information interprets the findings. So it has to be moved to the Discussion.

p23 Are all the diagrams necessary? Criterion: To which extent are they covered by the text?

p24 11- p25, line 5 All this information interprets the findings. So it has to be moved to the Discussion.

p25 19 - Reword.

- rw: reached -> crossed?

Fig. 9 - Merge with Fig 8.

p26 6-11 - All this information interprets the findings. So it has to be moved to the Discussion.

16-20 - All this information interprets the findings. So it has to be moved to the Discussion.

p27 It is hard to read out information from figure 10c To much included into a single diagram. Simplify!

p28 I am not sure whether this is essential to the paper's scope . . . I see some potential to shorten the paper by moving this to some Appendix.

p29 20 - rm: 'by a factor ..' Do not repeat results already reported on earlier. Instead, conduct a more clear-cut write-up of the obtained results.

- This gives a minimum age (only). And only for a strong phase that has never been stronger afterwards. That is, the meandering may have been triggered at an earlier point in time, but the pertinent strata was just cannibalised by the denoted activity.

22-23 - A strong statement. Still, is it actually supported by the calculated data given the inherent uncertainties? What if sediment transport rates (?quantity per unit time) was constant from an earlier time on? Isn't it possibly the same phenomenon as with terminal moraines? The most distal ones mark the last phase immediately before the 'dynamics' decreased. So they mark the onset of the decline. See the all the diagrams from 8 to 10, they all suggest a progressively declining meander activity.

p30 5-9 - Is perceived as speculative. Remove.

16-23 - Only speculation. Remove it, it is not connected to anything based on your methodology. Also, using climate data from the current climate normal carries a strong signal of climate change with characteristics being different from the pre-1980 period. The relationships that are constructed here are, therefore, very questionable.

Section 5.2.2 - Interesting, but how does it immediately relate to the methodology that was used? All the information is good for is to point out future avenues of research to clarify causes of what you observed on the floodplain (only). So this section should be shortened, dissolved, and merged with the hints that can be made regarding the role of post-Middle Ages climate fluctuations.

p32 14-26 - All of this only repeats content of the Introduction. Actually, there it was used to justify the research undertaken. But its occurrence in the context of the Discussion section means that is an outcome of the study? Delete the section.

27-33 - . . . and therefore this para should be part of the Intro. There it would add to provide a logical flow of justifying the research question in view of previous research.

Section 5.5. River management and restoration This section mostly reiterates commonplaces about fluvial morphology and stream restoration works. If you would like to keep it, then thoroughly rewrite it by making connections between your own findings and what they'd mean for the management and/or restoration efferts mentioned in section 2 ( case-based!). And include a the pertinent background to that in the Introduction. This topic is actually adding significance to the present research, even though the methodological approach as such is not necessarily novel. Try to link your research to the current debate on the meaning of 'natural rivers' and stream restoration goals (e.g., Brown et al., 2018, ESR).

p33 The 'Conclusions' - Are no true conclusions but yet another summary of the main findings. Moreover, what was discussed as possible causes and mechanisms in the previous section now is phrased as it was an evidence-based outcome of the study. Here, another complete rewrite was required.

Reduce # of in-text citations (adding too many citations does not add credibility): p2 - 8, 15, 17, 29, 33 p3 - 9 p6 - 15 p8 - 18 p30 - 32 p31 - 7, 15, 20 p32 - 19

Peter Houben Leiden University College

---

## Author Comment (AC1) · 12 Jul 2018

*I read a manuscript in a very well defined shape. The language, organisation, amount of references and overall quality is high and if the figures can be polished/optimised, the technical part will be of very good quality. I consider the topic of the study timely, relevant and well placed in the scope of the journal. The methods appear to be mostly adequate and thoroughly described. I especially welcome the general attempt to account for the uncertainty inherent to several of the inferred parameters, although there are further uncertainties that should be added to reach a more comprehensive capture of the total model uncertainty.*

Many thanks for your kind words and enthusiasm about the manuscript, and your critical review, adding very valuable suggestions. We will re-evaluate the chosen uncertainty for the parameters and add additional uncertainties where possible. Please find these details further below.

*Technically, I find the use of dotted and dashed lines in many of the figures disturbing. They make it sometimes very difficult to actually see the data that is to be visualised. For example in fig. 10C the dashed lines obscure the course of the data almost completely. Please think of reworking most of the dotted and dashed lines. In many cases they are not needed to make a distinction in the plots.*

We agree and changed figures according to suggestions.

*I strongly encourage the author(s) to provide along with the study also the code and data they used to generate their results. This would make it possible to reproduce their work and also increase the impact of the study. I have not doubt about the validity, rigour and correctness of the material but without seeing it I can hardly judge these points. Beyond that, readers of the paper will be happy to already have a starting point to proceed with if the code and data were presented along with the article.*

We will add a sheet with all the calculations that have been done, so that the reader can start very fast from there with their own calculations. We will also include our own used data in the spreadsheet as an example and a verification.

*I had the impression that there are some sections that are too inflated with information and detail, much more than what is actually needed to support the statements they are about to make. For example, the study area section, especially the restoration part, is interesting to read but very detailed, as well. Please consider restricting the content to what you essentially need to support your methods and the subsequent discussion/interpretations. Likewise, there are results reported in great detail that are not used to a reasonable extent, any more. The classic example for this is section 4.1. Such details may become part of supplementary materials but unless you need this for the discussion, it is not needed in the scope of the manuscript.*

We removed the section on restoration from the second chapter, also suggested by the other reviewer. Lithogenetic interpretation was moved into a table (as suggested by the other reviewer), including most important details. The manuscript has been shortened by ca. 3500 words, removing all repetitions, abundant results and discussions, in agreement with the three reviews.

*The abstract is mostly clear and gives a good overview of the topic and the main findings and their interpretations. It should however shed some more light onto the most detailed part of the study: the development and application of the calculus to describe hydrologic parameters and channel metrics.*

We added more detail on the reconstruction of the hydrological parameters and rephrased the potential causes part of the little ice age and peat reclamation.

*The introduction is well organised and follows a consistent flow of context. The references might imply that it is almost exclusively Dutch scientists that have worked on that topic. If that is the case, fine. If not, it might be good to also present adequate references from other regions. But this is just a suggestion that may help improving the manuscript.*

There are indeed some examples of Dutch cases (De Moor, Vandenberghe, Kasse, Hobo), but also many examples of non-dutch case studies in the 3th and 4th paragraph (Lewin, Slowik, Lespez, Notebaert and Verstraeten, Hoffmann, Kondolf, Piegay, etc.)

*The scope of the study as expressed at the end of the introduction is not a good match with what I read later on. The actual study goes way beyond the short summary of "detecting channel pattern change" and "identifying causes". Please give more details about the approach, as well. The field and especially the numeric work is a considerable and innovative part of your work and should be reflected by the scope definition.*

We sharpened the aim and focused more on the methodology of the palaeohydrological reconstruction in the introduction. We removed the aim of identifying the causes, but will only shortly elaborate on the potential causes in the discussion. So we put the focus more on the reconstruction than on the identification of the causes.

*Concerning the second part of the scope ("identifying causes"), this part is not ideally resolved, neither by your data nor by the discussion. In the latter part, you mainly cite other people's work and make a proposition that the Little Ice Age meteorological conditions and/or land use changes have had an influence on the observed/modelled results. But you do not and cannot easily go beyond this general statement. So maybe this part of the scope should not be a central goal?*

We agree and changed the scope, also in line with the other reviewers. We now focus more on the identification of the channel pattern change and methodology. See previous comment

*The study area description is fine, though in parts a bit too long. Please see detailed comments.*

We removed the section on river restoration from the study area description.

*The field methods description is in most cases conclusive and well understandable. See detailed comments below for some adjustments.*

Thanks

*The calculus description is less consistent. I acknowledge the idea of accounting for parameter uncertainty. But this must be done comprehensively and with justification. For several parameters there are either no uncertainties given or they appear out of the blue. See details below.*

We changed this and gave a better reasoning for each parameter on its uncertainty in the method section.

*The order of the equations does not match the order in which the text refers to them. So either re-order the equations or tweak the text to match the equations.*

Checked and changed

*The Chezy coefficient was assumed/estimated by several approaches. This is fine but in the discussion the average of all these approaches was used as the most likely value. I do not see a justification for this attempt. Are all these approaches equally likely or equally valid? If not, how and why was the final average coefficient value estimated?*

We agree and we changed the approach. In fact, Brownlie uses variables that are known, and of which we can vary the uncertainty. However, Manning is a subjective estimation of what the river looked like in the past. We changed the approach and only use the Brownlie, and we will compare the calculated Chézy value with values known from rivers of similar size and with similar river pattern.

*Overall, sections 3.7 – 3.11 introduce a large set of assumptions and equations/models. These are not well reflected in the introduction and scope of the manuscript. So, do you really need all these models to make your points and interpretations, or the other way around, are your research questions adequately addressed in the beginning to prompt such a large set of concepts and models?*

We understand the confusion, thanks. We changed the research questions accordingly. In fact, after we have identified the channel pattern change, we identify which parameters have changed, and we used the empirical models to test whether they can explain the channel pattern change.

*The set of parameter values were sampled and computed 200 times in a Monte Carlo approach. Are you sure that 200 MCMC runs are enough to cover the effects of variability adequately enough? From my experience with models that contain way less parameters I always needed much longer Markov chains to reach stable uncertainty estimates. Can you show that 200 is ok? Or have a test of convergence with number of model runs?*

We checked this, and raised the computed runs to 10.000 times.

*The results are mostly well presented. However, section 4.1 gives a very detailed picture of the lithology that is not used later to an extent that would justify this detail. I suggest to move this section to the supplementary materials to keep the story of the manuscript tight enough to be followed easily. Alternatively, make better use of the details in the discussion.*

Lithogenetic interpretation was moved into a table (as suggested by the other reviewer), including most important details. This section is important, because it is the fundament of the palaeohydrological reconstruction in which the palaeodimensions are derived from the cross-sections. Hence, the interpretation of the lithogenetic units is an important element in the manuscript

*The discussion sections should be reorganised to be more logical. I suggest to focus on time and not necessarily flow of context. You can/should start with the "laterally stable phase", then "channel pattern change", then "meandering phase", then "channel pattern reorganisation". This would keep the chain of information much more concise. You can implement sections like 5.2 into this system. I would also suggest to shorten section 5.5 considerably and have it as a conclusion theme. See details below.*

We agree and followed the suggestions. We reorganised the discussion according to the suggestions. We included the " channel pattern reorganisation" into the meandering phase. We removed section 5.5.

*Sections 5.2.1 and 5.2.2 are very detailed but mainly bring together findings from other studies, focusing on potential impacts of climate change and land use change. Please shorten and condense it to what you actually need to support your findings. It would be much more appropriate to have these*

*two sections organised together with section 5.2 (causes of channel pattern change) but also to make more links to your actual results. Actually, it is not really possible to disentangle the effects of "Little Ice Age weather" and "land use change" from your data situation. It can be either or both that may have drive your system of channel pattern change. Please mention this issue. It is no problem to have the effect of both.*

Agree, and merged this part with section 5.2. In addition, we shortened this part. See previous comment

*P 1, l 13-14, "changes in climate or land cover". There are certainly more that just these two drivers that can lead to changes in a regime. Consider changing to " changes in, for example, climate or land cover".*

We followed the suggestion by Referee#3, adding and/or.

*P1, l 17, "proven" is not a good term in the scientific approach. Consider replacing by "constrained".*

Agree and changed

*P2, l 20, consider changing "are documented of channel pattern changes" to "of channel pattern changes are documented".*

Agree and changed

*P6, l 6-7, hard to understand the value assignments. Consider rewriting to "with an average annular discharge Qm of 22.8 m3 /s and a mean annual flood discharge Qmaf or 160 m3 /s".*

Agree and changed

*P6, l 15-22, too detailed. Consider shortening significantly to an extent that matches the scope of the study.*

Agree and shortened

*P6, l 31-35. Actually all you can say is that the cutoff happened before 1720 AD. There is no information that supports the statements like "shortly before" or "date from the same period". Consider rewriting to stay with the available constraints.*

Agree, in this phase of the manuscript we can only take conclusions from its dimensions, but indeed not from the age. The study is needed to investigate this. We rewrote this section.

*P7, l 23, check overall the journal's definition of figure reference format rules, i.e., if "(Fig. 1(c)-(d))" is the right way.*

Checked and changed

*P8, l 6. The use of "respectively" makes it very hard in this sentence to identify the cases in which you used which device. Please rewrite like "In case we we used this device. In case B we use that device".*

Agree and changed

*P8, l 12, did I read this correctly, that you sieved material from a 3 cm wide auger/corer to estimate the gravel content? Is this a representative sample size, or in other words, over which depth interval did you have to average to get sufficient material for sieving?*

The sieving was meant to make an estimation for the gravel content in the lithological description. Purely meant as a field-based method to make a fast estimation, sufficient for the aim of this method: distinguishing the lithogenetic units

*P8, l 28, Add manufacturer info to grain size device (Beckman Coulter, Malvern, Horiba, etc.) to make clear which device you used.*

Changed

*P8, l 29. Check units. Is it 2000 m or μm?*

Checked, 2 mm is correct

*P8, l 30. Why was the Fraunhofer model valid? Was it "just" sandy material with minimum clay content? If not, the Mie model might be more appropriate.*

Yes, almost all sand, see Fig. 6. We changed the text slightly to make this more clear

*P8, l 33-34. Consider rewriting to simplify. E.g., "We used the scroll bar OSL burial ages determined by Quik and Wallinga. For details on the method see this reference."*

Changed according to suggestions

*P9, l 16, what is the consequence of the different age determination procedure for the palaeo channels? Are the Baysian constrained ages comparable to unconstrained ages? Are just the errors larger?*

Details for the age estimation and effects of Bayesian constraining are provided in Quik and Wallinga (https://doi.org/10.5194/esurf-2018-30). Some of the younger deposits are particularly poorly-bleached, and for those the cartographic evidence is leading and provided more accurate and precise ages. The Bayesian procedure overall resulted in smaller uncertainties..

We removed this sentence "apart from the final Bayesian.... from historical maps", because we already mention that only the laboratory analysis followed the same procedure. Their Bayesian analysis was a post-laboratory calculation. We added a short explanation why we did not use Bayesian analysis.

*P9, l 18, how were the radio carbon samples taken? From a corer or a pit?*

Changed, we used a piston corer, forgot to mention.

*P9, l 21, add HCl concentration*

Changed

*P9, l 31, Why did you assume a standard deviation of 5 %? Why this value? Does this come from the uncertainty arising from the GPR results? It should at least be justified somehow. Otherwise I could ask, why was it not assumed to be 0.5 % or 50 %?*

We reviewed this assumption. We introduced a standard deviation based on different assumptions for the channel dimensions (see comment to other reviewers), by 1) introducing two knickpoints and 2) determining it for both palaeochannels. Then we calculated the average and standard deviation of Hbf. Consequently, this approach also affects the other channel determined dimensions (A, P, R, W).

*P10, l 9, same as above, why the 5 %? Can you say something beyond "expert judgement"? It would considerably improve the impact and value of the study and since there are quite large uncertainty ranges in some of your results these input uncertainties may be crucial to evaluate the results. You can for example also think of sensitivity analysis. What would happen if you set the standard deviation to 1 % or 15 %?*

We agree and changed the assumption. We used the differences in surface and bottom elevation as a measure for the uncertainty of $H_{bf}$. See previous comment.

*P14, l 13. Is there any uncertainty available for the porosity value? Can you estimate a plausible value?*

We included an uncertainty range for the porosity of sand based on literature

*P14, l 14. Is there any uncertainty available for the age differences? Yes there is. So this should be included in the MCMC approach.*

There indeed is an uncertainty in the ages as shown by Quik & Wallinga (submitted), but the order of development of the scrolls in the scroll-bar sequence is known. Here we are primarily interested in the trend of the palaeodischarge over time, in particular comparison of the palaeohydrological conditions that existed at the start of the meandering phase.

The age estimate uncertainties are relevant for comparing the reconstruction to possible drivers, and are considered in our (condensed) discussion of these. Including these uncertainties in the figures showing trends in time is extremely complex, and if possible, would resulting in a blurred picture masking trends over time. Therefore we choose not to include the uncertainty of the ages.

*P14, l 24-25, the sentence does not fit very well, here. Consider shifting it to a more appropriate place where it does not cause a break in context.*

We moved the sentence a few sentences down.

*P15, l 6, Is there any uncertainty available for n? Can you justify why you chose 0.028 for this parameter?*

We found that taking a range of the Manning coefficient, the uncertainty becomes so high that it's rather useless. In fact, estimating the Manning coefficient was a matter of estimating what the river could have looked like (vegetation, irregularity) and comparing it with similar rivers, but this information is unknown. Based on reviewer comments, we have decided to only use the Brownlie formula, because it includes known variables and their uncertainty. We now compare the calculated Chézy with literature values.

*P15, l 12, Which type of rivers were these 79? sand bed? low land? Some detail is needed to understand the validity of averaging over this number of rivers.*

We moved this comparison to the discussion and added more detail on the rivers for which we averaged the Chézy value.

*P15, l 15, Who was the expert that suggested the value of the Chezy coefficient?*

Sentence was removed, because no details were given on how they estimated the Chézy.

*P15, l 24-25, give uncertainty estimate for intermittency and porosity parameters. Or say there is no uncertainty.*

We added the uncertainty for the porosity based on Nimmo's work. For the intermittency there is no uncertainty.

*P15, l 27, consider new paragraph between "available" and "In the second".*

Agree and changed

*P16, l 8-14. This is vital information about the stability diagram. Please deliver this earlier to the reader, e.g., when you first mention this diagram type. What is meant by "interpreted as a lower threshold, rather than a hard threshold"?*

We agree that the stability diagram is an essential part of the reconstruction. However, we refer in the introduction to the use of empirical channel models, which we further elaborate in the methodology section (here). We decided not to move this section to the introduction as it would make the introduction too long and unbalanced.

*P21, l 12, define or quantify the term "very similar", you have the data to do so.*

Changed

*P21, l 17, define or quantify the term "extremely slow", you have the data to do so. Also, you can make use of the uncertainty information.*

Changed

*P22, l 8, provide uncertainty information for slope of X.*

Changed

*P22, l 11, provide uncertainty information for slope of Q.*

Changed

*P22, l 11, what mean "relatively linear"? You should test and quantify. Actually I could also interpret a piecewise linear model with a break around 1850.*

We removed the comment

*P25, l 8-10, why did you choose the "middle of the ranges" and what is the "middle"? See above, why should the full range of estimated values for C be equally valid or likely? If they were equal, why would you make a distinction between "all rivers", "rivers without bars" and so on? Why did you use a standard deviation of 2 units? Please justify these apparently arbitrary assumptions. It is fine to include uncertainties, but their foundations must be reasonable.*

We removed this comment as Chezy is now based on the Brownlie equation, we don't use the other approaches anymore. See previous comments.

*P25, l 12, the values 32 and 38 are really really hard to map out on figure 9 a. And anyway does not everything in this figure drown in the uncertainty polygon? Please discuss your values with respect to the large uncertainty range.*

We added some discussion on the uncertainty in the graphs in section 4.4. An equal bankfull discharge would mean a large change of the parameters for Palaeochannel X&Q, which are unrealistic.

*P26, l 7, "was probably limited". . . not necessarily. You simply cannot resolve this statement with your data. Just that the phenomenon could be explained with option A (sediment transport is higher than bar growth) does not mean that option B (external sediment input) is not also contributing. Or would these two options be mutually exclusive?*

Removed this statement

*P29, l 7-14, this part contains very limited information but instead many repetitions of already discussed material.*

We rewrote this section. The repetition is caused because most of the discussion was already written in the results, therefore we moved it from the results into this section.

*P29, l 19-28, there is a lot of general information and unknown statements in this section. Please make a better connection to the results section. You have a lot of quantitative results, so please use them to support the statements made, here.*

We agree and rewrote this section.

*P31, l 31-32, if this is no tin the scope of the paper, then why referring to this topic?*

Removed this phrase.

*P32, l 11-12, I do not think it is actually possible to resolve whether or not increased sediment input played a role, so I would not mention this, here. See comment somewhere above.*

Removed this line

*P32, l 14-33, very broad and general. The main point I read from this paragraph is that we need more detailed field studies to pursue the question. Try to make more out of this material. It would be a valid goal to investigate if the one case you found in your study is an "outlier" or the "regular case". Anyhow, the paragraph in its current shape does not present/discuss your results. You have to make a story out of it or leave it.*

We removed it to prevent repetition, and discussed this part in the introduction.

*Likewise, the second paragraph comes a bit out of the blue. How does the Geul river come into play and why does it come into play, here? This section needs more context or should be skipped. Currently, it does not really match the section header.*

We moved this section to the introduction as suggested by the other reviewer, where it supports the likelihood that more rivers changed from laterally stable to meandering during the Holocene.

*P33, l 5-15, this part is also very broad/general and arm waving. Consider shortening significantly and link it much better to your concrete findings, i.e., what your case study can contribute to this*

*overall picture. Overall, I suggest to shorten this part and have it rather a conclusion item than part of the discussion.*

We removed this section on stream restoration to shorten the length of the manuscript and to keep the focus.

*P33, l 29, change "discharge increased" to "discharge potential increased".*

We rewrote the conclusion

*P33, l 30, change "exploitation has contributed" to "exploitation has probably contributed".*

Rewrote the conclusion

*Table 1, It would be better to have the radio carbon and OSL ages at the same scale. This concerns both, years versus kilo years and AD years versus absolute years. At the moment things are hard to bring together.*

Final ages are all presented in the same framework; following the revised manuscript of Quik & Wallinga (in press) we adopted the CE framework for this. Intermediate results are also presented reported in the appropriate unit, for the OSL ka ages are presented in addition to CE, as these relate directly to the reported palaeodose and dose-rate.

*Figure 1, replace dashed lines in panel b and f. Also, consider using solid lines to illustrate the zoom from panel to panel. Add similar "zoom lines" also from b to c and b to d. Provide a solid or at least partly transparent background to legends. The legend contents are really hard to see. Add legend frame in panel b.*

We changed the dashed lines into solid lines and also added them to panel b to c and b to d. The legend is poorly readable due to the low quality of the images. We decided not to add a background, because detail would get lost of the meander bend surroundings, also when transparent the surroundings will be hardly visible. However, the higher quality of the images improves the readability significantly.

*Figure 2, image quality is not good. Either this is due to the manuscript stage compression or other. It would be essential to add a higher resolved image of the GPR output. Also, the thick yellow lines are masking the raw data too much. Figure 5 does a much better job by showing both, raw and interpreted options. Alternatively, think of using thinner and semi-transparent lines. In figure caption there is repetition with "modified after Huisink" and "adapted after Huisink".*

We agree that the image quality is not optimal. This is because Huisink had a low quality image in her article, which is very likely due to the quality of the GPR output 18 years ago. Here the main goal is to illustrate the different subsurface features (palaeochannel, coversand, fluvioperiglacial) and showing that a symmetrical palaeochannel is present, but lateral accretion surfaces are lacking. This interpretation was not done by ourselves, but by Huisink already. For the actual data we refer to their work. For our own data (Fig. 5) we agree that it is essential to deliver good quality GPR images.

We changed the caption so that it becomes more clear that the interpretation was done by Huisink.

*Figure 4, Please decrease the size of the drawings and have all of them on one page. The context density of the drawings is not too high, you can scale them smaller without loosing much of the*

*content. Of course the axes labels and plot drawing texts must be rescaled to an appropriate font size. But currently, there is a mismatch in the size of the figures with respect to what they tell.*

The delineation of the scroll bar deposits and palaeochannel is an essential step in the reconstruction, and should be fully visible in Figure 4. We tried making the drawings smaller (so they would fit on one page), but too much detail was lost. We shortened the lithogenetic interpretation, therefore these figures become even more important.

*Figure 7, what do the errors want to tell in panels c and d? Overall, the resolution of the images are not really great Consider saving such plots are EPS vector data.*

We improved the quality of the images. The caption explains the uncertainty shown in Fig. 7c,d. This is the standard deviation of the Bayesian deposition model determined by Quik and Wallinga.

*Figure 8, figure quality/resolution is bad. Please avoid the dashed and dotted lines (e.g., panel g), they make it hard to see the data clean. Shift legend from panel a to panel c and d.*

We changed the resolution of the figures. We removed fig. 8gh, because they were not that important for the reader. We keep the legend in panel a, because here there is sufficient space, and the legend immediately explains the lines in panel a.

*Figure 10, dashed lines make it hard to see any trends*

Removed and replaced the dashed lines. We also added a log-plot to see more detail.

*Figure 11, why is Prathoek missing in above panel?*

As explained in the method section, we merged both meander bends together, because the same discharge and streampower are expected. For the IP this is different, because the IP is determined by channel-dependent parameters.

---

## Author Comment (AC2) · 12 Jul 2018

*The manuscript "Late Holocene channel pattern change from laterally stable to meandering caused by climate and land use changes" aims to identify river channel pattern changes using sedimentary and geochronological data and to identify causes for these changes. The manuscript is well written, the topic is relevant and in the scope of the journal, and the concepts and ideas are sufficiently novel. The methods are consistent and well described. There are some minor to moderate shortcomings, listed below. When these shortcomings are resolved, I consider this manuscript as a valuable contribution to Esurf.*

Thanks for your kind words and critical review adding significant value to the manuscript. Below we respond to each of the suggested changes.

*- Some sections are written too extensively, and not all information is needed to answer the research questions. For instance, the details on river restoration in section 2 are not needed and can be limited to a minimum. Also section 5.4 and 5.5 can be shortened.*

The same suggestions were given by Referee #1. We shortened or removed the suggested sections or merged them together. Our manuscript was shortened by ca. 3500 words, removing repetition and excessive information.

*Section 2 (study area): P6, L29-37: A lot of assumption are made in this part. I suggest to move this part to section 4.2 (results). And then in section 4.2, you have to provide all available arguments to state that channel X is predating the meandering phase. Show data to support your statements (eg show the GPR profile). You have to provide good arguments to state that channel X is from a laterally stable phase, since this is an important point for the rest of the story.*

We changed this section, removing the assumptions, as also proposed by reviewer #1. We indeed have no information on the age or stability yet, that's one of the outcomes of this research. We leave the introduction of palaeochannel X in this section, so our steps in the methods become more clear why we are investigating this palaeochannel.

*Section 3.1 is not needed to my opinion. Aims are already explained in section 1 (Introduction); methods will be described in detail in the next paragraphs (3.2 and next sections).*

Agree and changed also in according to the suggestions by the other reviewers.

*- P9, line 29: How did you define the knick-point on the bank? What will be the effect on bankfull depth and discharge when using a different knick-point on the bank? You can try a sensitivity analysis to check the effect of the definition of the bankfull depth.*

We measured in high resolution the banks of the palaeochannel with a GNSS. However, we now introduce uncertainty by taking the first clear knick-point on both banks, causing differences in channel dimensions.

*P9, line 31: Why a standard deviation of 5%? Which arguments do you have? This is an important point, since large parts of your interpretations are based on this standard deviation. If you assume a standard deviation of 10 or 20%, it is possible that your differences explained in figure 8 are not so clear anymore. Can you provide a consistent method to define the standard deviation? Also here, you*

*can try a sensitivity analysis to check the effect of the standard deviation. - Same question for P10, line 9.*

We reviewed this assumption. We introduced a standard deviation based on different assumptions for the channel dimensions (see comment to other reviewers), by determining the relative error of Hbf for the meandering phase and assuming a similar relative error for the laterally stable phase, because both estimates are based on coring data. The relative error is ca. 10% of the Hbf. We took the same percentage of relative error for the other determined channel dimensions (A, P, W).

*– Section 4.1: You can summarize this section in a table showing the most important characteristics of the different lithogenetic units. The table can then be followed by a short paragraph on defining the scroll bars and scroll bar dimensions.*

Agree and changed according to your suggestions. The other reviewers also agreed that section 4.1 should be shortened, hence a table provides a good solution to do so.

*Section 4.4: L11: Use statistical tests to check if the reconstructed discharge differs significantly. Given the uncertainty range it is possible that you can not reject the null hypothesis (Q does not differ). The same for L13: 'Q drops relatively fast at 1800 AD': Given the uncertainties, it is possible that Q is not significantly different. Use statistical tests to support your statements. P 29, L20: It is also likely that the discharge does not differ significantly, given the uncertainties. See my previous comment.*

We added a section on how much parameters have to change to reach similar results in sect. 4.4. These factors fall outside the range of the uncertainty of these parameters, hence values between the laterally stable phase and meandering phase are significantly different.

*- Section 5.2: This section mainly brings together results of previous studies and it is not based on new data. So this section should be shortened and should link better to your own data and findings. Try to better link quantitative data on climate change and land use changes with your findings.*

Agree, we rewrote this section and merged it with 5.2.1 and 5.2.2.

*- Section 5.2.2: Is there an observed increasing in urbanization in your catchment? Urbanization can cause higher peak discharge, which have been described in catchments in The Netherlands.*

Urbanization is an important factor in recent land use changes during the last century, where paved roads cause flood increases. However, during the Middle Ages roads and cities were by far not that well developed compared to recent developments (see also Lanen et al., 2015 on archeological studies on road infrastructure, including the area of interest)

*- P31, L6 and L11: 27% of the catchment was covered with peat + yearly average discharges can increase by 40% => ca. 11% increase in average discharge for the entire catchment. How does this compare to your reconstructed increase in discharge?*

Thanks, we added this comparison to the text.

*- P 31, L 29-31: "Our data strongly suggest": not correct. As you stated in section 5.2 it is likely that the increasing discharge caused the change; you have some good suggestions but no hard evidence.*

Changed the entire section according to suggestions of reviewer 1. We removed these strong statements.

*"The most likely identified causes": actually these are the only factors checked. You did not checked other contributing factors. –*

Changed the text and the aim of the paper in agreement with all reviewers. Identifying the causes is not the main aim of the manuscript anymore, hence we mention the likely causes.

*Figure 4: Indicate the location of the datings on Figure 4e.*

We added the locations of the datings on Figure 4e and extended the southern part of the figure so they would all be included.

*- Figure 10c: this figure is not entirely clear. The dashed lines do not help. Try to simplify this graph to make it more clear.*

Changed the figure according to suggestions. We removed the dashed lines and added a log-scale to the y-axis to make everything more visible.

*- References: For some references, correct volume, issue and pages are missing: P36, L5-6; P36, L24-26; P36, L56-57; P37, L40-41 (I may have missed more).*

Check and changed

---

## Author Comment (AC3) · 12 Jul 2018

*The paper on "Late Holocene channel pattern change [. . .]" by Candel et al. reports*
*on the use of floodplain stratagr. records and chronologies to conduct a quantitative*
*assessment of (paleo)hydrological channel planform change over the past 600 years*
*in the NE NL. At a general level the manuscript is organised, the introductory sections*
*provide background to the research, and give sufficient detail of the used methodology.*
*The methodological approach and the subsequent evaluation of obtained results are*
*based on a strong research effort, and the discussion puts the work in the context*
*of previous work and addresses potential implications. All of which fits the journal's scope.*
*All in all, the ms represented a valuable contribution to ESurfD, however, in its current*
*form it requires restructuring and (partially) rewriting at the paragraph level. Regarding*
*given standards a number of statements are misplaced. For example, the Results*
*section includes discussions of the findings, which is why the actual Discussion mostly*
*reverts to a sometimes narrative analysis. The weakest sections, thus, are the Discussion*
*and the Conclusions wherein some thoughts brought up and connections that are*
*sought to be made should be reconsidered with respect to whether they actually add to*
*the paper's significance. In consequence, the abstract should be rewritten because it is*
*not reflecting the actual paper content (and the balance of the featured aspects), and*
*the highlighted findings are not supported by the employed methodology. At places*
*abundant in-text citations in the Introduction can be perceived as a bit too excessive.*
*Key: rm - remove; rw - rewrite/reword;*

Thanks for your kind words and your critical and valuable review on the manuscript. We agree and are
thankful that your review made clear that the structure of the text should be improved. We moved
large parts of the results to the discussion and focussed the discussion more on the main findings. Also
we rewrote the conclusion and abstract.

*p1: Title: Actually, the paper does not include hard information that allows for pointing*
*to the actual causes of the described channel change. In the paper, a number of (truly)*
*possible and plausible causes are mentioned but no conclusive evidence can be shown*
*that helped to causally link channel change to either or both of the drivers. Why not*
*highlighting the strength of the paper, the application of quantitative palaeohydrological*
*approaches to answer the actual research question?*

With the new input of the reviewers we agree that the title does not match the content anymore.
Therefore we changed the title accordingly, highlighting both the channel pattern change and the
palaeohydrological reconstruction.

*13 - The Abstract . . . "related to changes in climate and/or land"*
Changed accordingly

*15-18 - Results are reported before the actual scope of the paper is given. And the*
*approach is only explained later on. Rearrange to present a logical flow.A*
Agree and we rearranged the order

*18 - Actually, no potential causes have been investigated. This is misleading information.*
*Only other people's work is cited in the Discussion when attempting to explain*
*what possible causes have been around. The nature of that discussion, nevertheless,*

*remains speculative.*
Removed the sentence

*28 -29 - 'reflecting relative . . .' this statement should be rephrased because it it ambiguous, and overall not intelligible when only reading the abstract.*
Changed

*31 - The last sentence is not specific to the paper content, rather will appear like a motherhood statement the the journal's audience. Remove and replace it by strong statements that stress the significance of the own findings. The reason for the weak end of the Abstract, my guess, is the underdeveloped Conclusions section (see below).*
Agree and changed

*34 - 'Several . . .' Sentence can be deleted.*
Removed

*p2 5 - In a braided river system, isn't the temporary presence of laterally stable/ migrating channels (runnels) just a matter of stage at a time?*
Here we make a statement that refers to laterally inactive rivers, and rivers that show lateral migration. Both meandering and braided rivers can have channel reaches that are temporary laterally stable. However, both meandering and braided rivers show in general laterally migrating channels. In this case the differences in processes between meandering and braiding rivers are irrelevant.

*7 - ' variables like potential . . .'*
This is a matter of taste; we prefer our phrasing and made no changes.

*7 - rm: ', which is . . . slope'*
This information is needed to understand why Qbf is reconstructed, which leads to the stream power. Therefore we leave this sentence.

*8 - '2011), bank erodibility (. . .), cohesiveness (. . .), and by vegetation (...).'*
Changed, but differently than suggested. Bank cohesiveness and vegetation are important factors determining the bank erodiblity, so they should not be equally summed up.

*9-10 - rm: 'which is . . . (Turowski, . . .)'; 'that can increase . . . '*
Changed, but differently than suggested. See previous response

*6 vs 11 - Statements contradict each other*
Changes above have solved this

*13 - rm: gradually*
Changed

*13-19 - shorten para*
The paragraph consists of vital information of our current state of knowledge on channel pattern changes, which is entirely based on the change between meandering and braiding planforms.

We removed a few references to shorten the paragraph as suggested later, but did not shorten the sentences.

*23 - rw: 'the exception is formed by human intervention'*
Changed

*23-34- This para does not fit in here. The surrounding text provides background information that should translate into the 'gap' and clearly formulated research goals,*

*however, this para explains processes of channel change. Could be moved together*
*with p2 10-18 to line 18 on p3.*
We moved this para together with the suggested para.

*33 - Excessive citing . . . Can the information be organised into a table?*
We decided to remove references and provide only references that refer to multiple river systems.

*p3 11 - It feels as if already here the paper's research question is addressed, but the*
*authors then return to reviewing literature.*
Removed last sentence of paragraph

*19 - Shouldn't the information be part of the first para on p3?*
We merged these lines together with the previous para
*20 - '.. stable channels poorly preserve except for . . .' rm all the rest between 21 and*
*25*
Changed, and the removed lines are left for the discussion

*31 - ' Huisink, 2000) while the meandering pattern has remained throughout . . .'*
Changed accordingly

*33 - rm: 'However'*
Removed

*p6 14-16 - This needs to be moved to the Intro. There, it was already used to justify the*
*research effort. In general, most of the content of p6 should be part f the Intro because*
*it is the background against which the present investigation can be justified. (I.e., it's*
*potential value to inform restoration projects.) This is even more important as this point*
*is picked up in the discussion s one of the more significant implications . . .*
Rather than moving it to the intro we removed this para, as suggested by the other reviewers.

*34 - In far can could the used features by local peculiarities due to their peculiar morphological*
*context?*
Unclear what is meant here, but we changed this section according to suggestions by the other
reviewers

*p7 7 - First sentences should not lead the Methods sections. Stating the paper goals*
*belongs to the Intro.*
We removed this section and stated this part more clear in the introduction
*7-21 - The whole para is a mix of review (again) and methods description. Needs to be*
*rectified.*
Removed and moved to introduction section

*Fig. 2, A - B - C designation is hardly readable.*
You refer to use of the Fig2(a) and 2(b) etc.? We changed this for all references to figures.

p8 6 - r: ' (i.e. the full ..)'
Removed

*11 - Estimating a statistical parameter for which others apply stacks of sieves by just*
*visual(?!) means? That might work depending on what the information is used for. For*
*me this is a point for of major concern. Actually, the D50 value is key to the calculations*
*performed employing eq. 8, 10, 12, 15, and 16.*
This data was only used for the lithological description. The grain size analysis was used for the D16,
D50 and D84, so the D50 in the equations was not based on the visual assessment. This is also
described in section 3.4. We added a sentence to make this more clear.

*While in general the methodology also accounts for ranges or error, I am not convinced that the 5% uncertainty is fair for this error-prone guesstimate. How good (=reliable, =reproducible!) can the far-reaching conclusions drawn be? (E.g., see fig. 10).*
We did not use a 5% uncertainty for the D50, but we used the standard deviation derived from the grain size analysis

*17-19 - rm: 'GPR . . . 2011).'*
Removed
*29 - replace: over -> with*
Changed
*30 - What sort of laboratory prescriptions (= 'instructions')? Sounds like voodoo science, doesn't it?*
Removed this additional statement, not needed.
*33- rm: the*
Changed according to suggestions of other reviewer
*33 - rm: 2nd sentence*
Changed according to suggestions of other reviewer
*35 - 'The scroll bars' . . . can be removed, or reword, or ..*
Changed according to suggestions of other reviewer
*p9 18 - rm: first sentence*
Changed
*21-22 rm: whole sentence, it's just nomeclature*
This definition is essential to determine where the sand-peat interface is located, and is necessary to report for the repeatability of the study. We decided to leave this sentence.

*31 - Why 5%? Can you justify this? Still a rather optimistic estimate.*
Agree, see previous response to other reviewers. We reviewed this assumption. We introduced a standard deviation based on different assumptions for the channel dimensions, by determining the relative error of Hbf for the meandering phase and assuming a similar relative error for the laterally stable phase, because both estimates are based on coring data. The relative error is ca. 10% of the Hbf. We took the same percentage of relative error for the other determined channel dimensions (A, P, W).

*p10 Insert space between Fig. 3 and the text. The figure even may be left out.*
Changed. We will leave the figure in, because it clarifies how the equations 1-3 were derived.

*p11, 12 Nice figures. However, would it work for people who printed it in B/W?*
*Checked, and changed the colours of the lithogenetic units slightly to assure that B/W print will work. For the lithological cross-sections these colours can be distinguished.*

*p14 29-32 - How was D16, D84 determined? Also visually? From the waterlogged sands that spread to either side when the sample material is pushed out of the Vander-Staay tube? I think this is a soft point of the methodology, in particulary with respect to the heavy mathwork that follows to nail physical, hydraulic parameters of in-channel water and sediment flow.*
See comment above. The D16, D50, D84 were derived by the grain size analysis, not by the visual assessment. We added a reference in this line to section 3.4 to make this more clear.

*p15 15 - State what was actually used here. Rather an issue of the methodology than a result.*
We removed this sentence and decided only to use Brownlie, as suggested by the other reviewers. In fact, Brownlie uses variables that are known, and of which we can vary the uncertainty. However, Manning is a subjective estimation of what the river looked like in the past. We changed the approach

and only use the Brownlie, and we will compare the calculated Chézy value with values known from rivers of similar size and with similar river pattern.

*27 - New para.*
Changed
*p16 28-33 - rm: 2nd sentence*
Removed

*p18 All in all, the whole Methods section could be more concise, focused. It would be worth to focus on the most important aspects and move the remainder to the Appendix.*
We applied all the suggested changes by the 3 reviewers making the methods more concise. Reviewer 1 and 2 are in general positive about the methodology section, therefore we decided not to move any section to the appendix.

*p19 6-11 - Reword.*
Partly rewritten
*10 - rm: 'Such a clear . . .Prathoek'*
Changed
*11 - rm: last sentence*
Changed
*20 - rw: abundant above*
Text has been removed in response to other reviewers
*Whole section 4.1.: Commonly, the ordering of geol. units is from old to young.*
Changed in the newly introduced table, and we shortened the text..
*p21 22-28 - 'Palaeochannel . . .' All this information interprets the findings. So it has to be moved to the Discussion.*
Moved to discussion where we discuss the laterally stable phase
*p23 Are all the diagrams necessary? Criterion: To which extent are they covered by the text?*
We removed figures 8gh and 9b, because they were not abundantly referred to in the text.
*p24 11- p25, line 5 All this information interprets the findings. So it has to be moved to the Discussion.*
Agree and moved to discussion
*p25 19 - Reword.*
Reworded.
*20 - rw: reached -> crossed?*
Changed
*Fig. 9 - Merge with Fig 8.*
Because we merged Junnerkoeland and Prathoek in the calculations for discharge and flow velocity, we won't merge Fig9 and 8, because in Fig. 8 they are still separated. In addition, Fig. 9 is important and deserves more attention, so it can better be separated.

*p26 6-11 - All this information interprets the findings. So it has to be moved to the Discussion.*
Moved to discussion

*16-20 - All this information interprets the findings. So it has to be moved to the Discussion.*
Moved to discussion

*p27 It is hard to read out information from figure 10c To much included into a single diagram. Simplify!*
We simplified the graph, mainly by removing the dashed lines and making the y-axis logarithmic.

*p28 I am not sure whether this is essential to the paper's scope . . . I see some potential to shorten the paper by moving this to some Appendix.*

This part is essential to our key message, because with these empirical models we test the likelihood that discharge increase has led to the channel pattern change. So can we use the palaeohydrological parameters (including their uncertainties) and understand why the channel pattern has changed to meandering.

*p29 20 - rm: 'by a factor ..' Do not repeat results already reported on earlier. Instead, conduct a more clear-cut write-up of the obtained results.*
Removed

*21 - This gives a minimum age (only). And only for a strong phase that has never been stronger afterwards. That is, the meandering may have been triggered at an earlier point in time, but the pertinent strata was just cannibalised by the denoted activity.*
Yes, we improved the argumentation for this point. If earlier, the meandering activity has not been preserved, but we would still expect to find more channel cut-offs or meander scars in the floodplain, or some older scroll bar deposit. Even when the new meander cannibalised the old one.

*22-23 - A strong statement. Still, is it actually supported by the calculated data given the inherent uncertainties? What if sediment transport rates (?quantity per unit time) was constant from an earlier time on? Isn't it possibly the same phenomenon as with terminal moraines? The most distal ones mark the last phase immediately before the 'dynamics' decreased. So they mark the onset of the decline. See the all the diagrams from 8 to 10, they all suggest a progressively declining meander activity.*
Removed the sentence. However, we reconstructed the sediment transport based on the reconstructed channel dimensions, so the actual sediment transport at that time. The scroll bar growth is not a lagged effect, but is determined by the actual amount of erosion and deposition, and hence the sediment availability. In this case, we refer to the moment of the channel pattern change, not to the decline during the meandering. Scroll bar growth can only be this high during the channel pattern change because of an increase in sediment transport, which has increased as a result of the discharge.

*p30 5-9 - Is perceived as speculative. Remove.*
Removed this section

*16-23 - Only speculation. Remove it, it is not connected to anything based on your methodology. Also, using climate data from the current climate normal carries a strong signal of climate change with characteristics being different from the pre-1980 period. The relationships that are constructed here are, therefore, very questionable.*
Removed this section

*Section 5.2.2 - Interesting, but how does it immediately relate to the methodology that was used? All the information is good for is to point out future avenues of research to clarify causes of what you observed on the floodplain (only). So this section should be shortened, dissolved, and merged with the hints that can be made regarding the role of post-Middle Ages climate fluctuations.*
Followed the recommendations, also in agreement with the other reviewers.

*p32 14-26 - All of this only repeats content of the Introduction. Actually, there it was used to justify the research undertaken. But its occurrence in the context of the Discussion section means that is an outcome of the study? Delete the section.*
Agree and deleted this section

*27-33 - . . . and therefore this para should be part of the Intro. There it would add to provide a logical flow of justifying the research question in view of previous research.*
Agree and moved to introduction to support the research goals.

*Section 5.5. River management and restoration This section mostly reiterates commonplaces*

*about fluvial morphology and stream restoration works. If you would like*
*to keep it, then thoroughly rewrite it by making connections between your own findings*
*and what they'd mean for the management and/or restoration efferts mentioned in*
*section 2 ( case-based!). And include a the pertinent background to that in the Introduction.*
*This topic is actually adding significance to the present research, even though*
*the methodological approach as such is not necessarily novel. Try to link your research*
*to the current debate on the meaning of 'natural rivers' and stream restoration goals*
*(e.g., Brown et al., 2018, ESR).*
We removed the section, also following suggestions by the other reviewers. In future work we will definitely discuss its relevance to river management and restoration.

*p33 The 'Conclusions' - Are no true conclusions but yet another summary of the main*
*findings. Moreover, what was discussed as possible causes and mechanisms in the*
*previous section now is phrased as it was an evidence-based outcome of the study.*
*Here, another complete rewrite was required.*
Rewrote the conclusions

*Reduce # of in-text citations (adding too many citations does not add credibility): p2 -*
*8, 15, 17, 29, 33 p3 - 9 p6 - 15 p8 - 18 p30 - 32 p31 - 7, 15, 20 p32 – 19*
Removed least important citations for the suggested locations.

Peter Houben Leiden University College